# Self-supervised learning reveals clinically relevant histomorphological patterns for therapeutic strategies in colon cancer

Bojing Liu[1,2,10], Meaghan Polack [3,10], Nicolas Coudray[2,4], Adalberto Claudio Quiros[5], Theodore Sakellaropoulos[2], Hortense Le [2], Afreen Karimkhan[6], Augustinus S. L. P. Crobach[7], J. Han J. M. van Krieken[8], Ke Yuan [5,9], Rob A. E. M. Tollenaar[3], Wilma E. Mesker[3] & Aristotelis Tsirigos [2,6] ✉

Self-supervised learning (SSL) automates the extraction and interpretation of histopathology features on unannotated hematoxylin-eosin-stained whole slide images (WSIs). We train an SSL Barlow Twins encoder on 435 colon adenocarcinoma WSIs from The Cancer Genome Atlas to extract features from small image patches (tiles). Leiden community detection groups tiles into histomorphological phenotype clusters (HPCs). HPC reproducibility and predictive ability for overall survival are confirmed in an independent clinical trial ($N = 1213$ WSIs). This unbiased atlas results in 47 HPCs displaying unique and shared clinically significant histomorphological traits, highlighting tissue type, quantity, and architecture, especially in the context of tumor stroma. Through in-depth analyses of these HPCs, including immune landscape and gene set enrichment analyses, and associations to clinical outcomes, we shine light on the factors influencing survival and responses to treatments of standard adjuvant chemotherapy and experimental therapies. Further exploration of HPCs may unveil additional insights and aid decision-making and personalized treatments for colon cancer patients.

Traditionally, the diagnosis of colon cancer is confirmed by microscopic assessment of resection specimens on hematoxylin and eosin-stained (H&E) slides by pathologists. For each patient, a personalized treatment strategy is tailored through a multidisciplinary meeting, following guidelines consisting of risk assessments based on clinicopathological characteristics, including the tumor-node-metastasis (TNM) classification and additional biomarkers[1-4]. However, due to an aging population and increasing amount of biomarker research, diagnosing and predicting the prognosis of colon cancer patients can be time consuming, or complicated and resource-demanding, especially when incorporating screening for mutational variants[2,3,5,6].

In modern digital pathology, scanning H&E slides into high-resolution whole slide images (WSIs) has enabled the applications of

[1]Department of Medical Epidemiology and Biostatistics, Karolinska Institutet, Karolinska, Sweden. [2]Applied Bioinformatics Laboratories, New York University Grossman School of Medicine, New York, NY, USA. [3]Department of Surgery, Leiden University Medical Center, Leiden, The Netherlands. [4]Department of Cell Biology, New York University Grossman School of Medicine, New York, NY, USA. [5]Department of Computing Science, University of Glasgow, Glasgow, United Kingdom. [6]Department of Pathology, New York University Grossman School of Medicine, New York, NY, USA. [7]Department of Pathology, Leiden University Medical Center, Leiden, The Netherlands. [8]Department of Pathology, Radboud University Medical Center, Nijmegen, The Netherlands. [9]School of Cancer Sciences, University of Glasgow, Glasgow, Scotland, UK. [10]These authors contributed equally: Bojing Liu, Meaghan Polack. ✉e-mail: Aristotelis.Tsirigos@nyulangone.org

deep learning (DL)[7]. Deep convolutional neural networks, in particular, have benefited the diagnostic process, initially by minimizing inter-rater disagreement and workload[7–9]. In colorectal cancer, supervised DL models also showed the ability to predict molecular pathways (i.e. mutation density, microsatellite instability [MSI], chromosomal instability) and key mutations like *BRAF* and *KRAS*[10,11]. DL even has intriguing potential in predicting complicated prognostic outcomes such as patient survival[12–14]. Moreover, integrating multi-omic data with the associated H&E slides, i.e. multimodal data integration, led to improvements in prognostic prediction on overall survival (OS) for most cancer types[15,16].

Previous DL studies primarily focused on training models to extract features from WSIs under supervision of potentially extensive and time-consuming human-derived annotations on slide or pixel level, i.e. supervised learning[10,12,15,17]. Self-supervised learning (SSL) on the other hand, has gained significantly increasing attention for its capacity to automatically capture image features from unlabeled data[11]. Applications of SSL models, including general-purpose foundation models[18,19], have demonstrated superior performance in various downstream cancer classification, survival, and molecular phenotype prediction tasks compared to traditional supervised learning models[20–24]. Barlow Twins, an SSL model designed to learn non-redundant image features, has several advantages over other SSL learning models (e.g. contrastive learning models), including not requiring extensive batch sizes nor asymmetry between the network "twins"[25].

Despite the efficacy in decision-making, DL models are often labeled as "black boxes", posing significant challenges in terms of interpretability. Supervised attention-based multiple-instance learning is a common interpretive method, enabling DL models to concentrate on informative segments within WSI according to predefined training labels[26]. Another approach involves employing unsupervised clustering algorithms to organize extracted features into clinically relevant and interpretable clusters which can be subsequently linked to diverse patient-related outcomes[23,27,28]. These clustering methods offer significant advantages, including the prediction of various clinical outcomes, intuitive visualization for pathologists, and interpretation and correlation with a range of molecular data.

In this work, our objective is twofold: first, to automatically and reliably extract clinically relevant histologic patterns from WSIs, which can be interpreted by pathologists, and second, to investigate the connections between these patterns and patient outcomes as well as molecular phenotypes across different treatment groups within a large clinical trial for colon cancer. To achieve this, we apply an SSL pipeline[23] involving the Barlow Twins encoder for feature extraction, followed by a community detection algorithm to construct an unbiased atlas of histomorphologic phenotype clusters (HPCs). This algorithm is exclusively trained on public data from the colon adeno-carcinoma cohort within The Cancer Genome Atlas (TCGA) multi-institutional database[29]. Remarkably, the identified HPCs generalizes well in unseen WSIs obtained from the clinical Bevacizumab-Avastin® adjuVANT (AVANT) trial[30]. Subsequently, HPCs are linked to patient OS. An HPC-based classifier trained using TCGA data for OS, demonstrates prognostic significance in the external validation of the AVANT study, even when considering key clinical and demographic factors typically employed in clinical settings. Notably, by conducting comprehensive analyses of the distinct histomorphologic features of each HPC and their associations with immune and genetic profiles, we provide insight into morphological and molecular determinants of patient survival upon different treatments (e.g. standard-of-care adjuvant chemotherapy and experimental targeted therapies). Exploring these features further could yield additional insights into other histopathology diagnostics, supporting shared decision-making and advancing personalized treatment options for colon cancer patients in the future.

## Results

### Self-supervised learning of WSI features using the multi-institutional TCGA dataset

We trained the self-supervised algorithm using data exclusively from the TCGA colon adenocarcinoma (TCGA-COAD) set, eliminating the need for annotations by pathologists (Fig. 1). A total of 435 WSIs (428 patients) obtained from the TCGA-COAD dataset (see Methods: Study population for details) were first divided into smaller image patches (224-by-224 pixels), also known as image tiles, at a magnification level of 10x (Fig. 1a) (see Methods: Data pre-processing for details). To identify features on these patches, we trained an SSL Barlow Twins feature extractor using a random subset of tiles ($N$ = 250,000 image tiles) (Fig. 1a) (see Methods: Extracting image features using Barlow Twins for details). The Barlow Twins was trained with the objective function to evaluate the cross-correlation matrix between the embeddings (feature vector $\mathbf{z}$) of two identical backbone networks, which were fed distorted variants of a batch of image tiles. The objective function was optimized by minimizing the deviation of the cross-correlation matrix from the identity matrix. This led to increased similarity among the embedding of $\mathbf{z}$ vectors of the distorted sample versions, while reducing redundancy among the individual components of these vectors. As a result, each tile was described as a vector of 128 extracted features that can subsequently be used to group tiles into clusters by similarity.

### Construction of an unbiased atlas of histologic patterns through community detection

We applied the Leiden community detection algorithm to derive HPCs, i.e. the clusters with similar histologic patterns (Fig. 1b) (see Methods: Identification of HPCs for details). The process began by first projecting the trained Barlow Twins onto the entire TCGA-COAD dataset, extracting 128 dimensional feature representations for each image tile. Subsequently, we utilized Leiden community detection on a nearest neighbor graph constructed from these tile vector representations (Fig. 1b). Tiles with similar vector representations were clustered into a group and assigned a specific HPC ID number. The optimization of the Leiden configuration was achieved through an unsupervised process (see Methods: Identification of HPCs for details, Supplementary Fig. 1a), resulting in the identification of a total of 47 HPCs, visually represented in a dimensionality reduction plot (Uniform Manifold Approximation and Projection; UMAP plot) (Fig. 2a).

As an external dataset, we analyzed a total of 1213 colon cancer patients with diagnostic pathology H&E WSIs (one WSI per patient), a subset of the clinical AVANT trial[30,31] (see Methods: Study population for details). We harnessed the optimized SSL Barlow Twins model to generate embeddings of the unseen AVANT WSI tiles. The assignment of identified HPCs to the unseen AVANT was achieved using the K-nearest neighbors approach. The HPC label of each tile in the AVANT data was determined based on majority votes from its K-nearest neighbors (K = 250) in the TCGA training set (Fig. 1b). As a result, we obtained comprehensive visual representations of the WSIs where the WSI tiles are colored by their corresponding HPC (Fig. 1b). Additionally, we were able to capture the characteristics and heterogeneity of WSIs using the compositional data derived from the HPCs, i.e. the percentage of the area on a WSI covered by each HPC, thus facilitating downstream analyses and modeling (Fig. 1c).

### Histopathological assessment and characterization of HPCs

Each HPC underwent histopathological analysis on a randomly selected set of 32 tiles per cluster within TCGA, independently evaluated by two pathologists (ASLPC and JHJMvK) and a researcher (MP)(see Methods: Interpretation of HPCs for details). Tissue types, as observed on the tiles, were described with specific attention to tumor epithelium, tumor stroma and immune cells. Other unique histopathological features or patterns, such as tumor differentiation grade and stromal

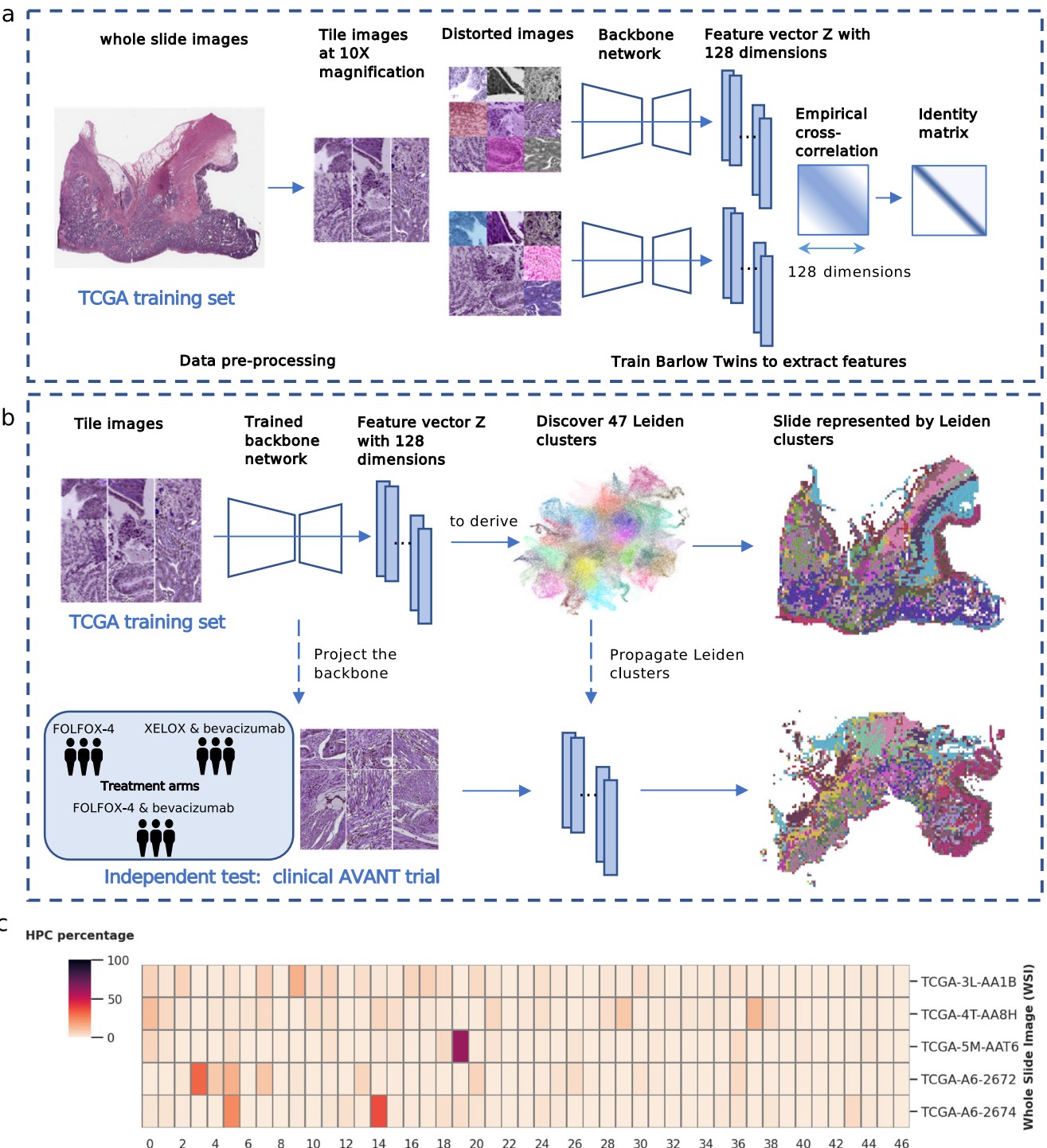

**Fig. 1 | Overview of the Model Architecture: Training Barlow Twins and deriving Histomorphological Phenotype Clusters. a** Training Barlow Twins with TCGA. WSIs from TCGA were processed to extract image tiles and normalize stain colors. The Barlow Twins network was employed to learn 128-dimensional **z** vectors from a random sample ($N = 250,000$ image tiles) of these image tiles. **b** Deriving HPCs. The tiles from TCGA were projected into **z** vector representations obtained from the trained Barlow Twins network. HPCs were defined by applying Leiden community detection to the nearest neighbor graph of **z** tile vector representations. Each WSI was represented by a compositional vector of the derived HPCs, indicating the percentage of each HPC with respect to the total tissue area. The Barlow Twins model and HPCs were then projected and integrated into the external AVANT trial. **c** Whole Slide Image Representation. The compositional HPC data represented the WSIs in the study. AVANT, Bevacizumab-Avastin® adjuVANT trial. HPC, histomorphological phenotype cluster. TCGA, The Cancer Genome Atlas. WSI, whole slide image. Source data are provided as a Source Data file.

organization, were noted as well. All present tissue types were scored in percentages (Supplementary Table 1) and depicted using pie charts (Fig. 2b). We plotted the interconnections of 47 HPCs using partition-based graph abstraction (PAGA)[32], with the pie charts reflecting their tissue compositions (Fig. 2b). Interestingly, distinct larger groups of clusters, or "super-clusters", could be observed based on the similarity of tissue composition, interconnectedness, and topology of HPCs in the PAGA plot (Fig. 2b).

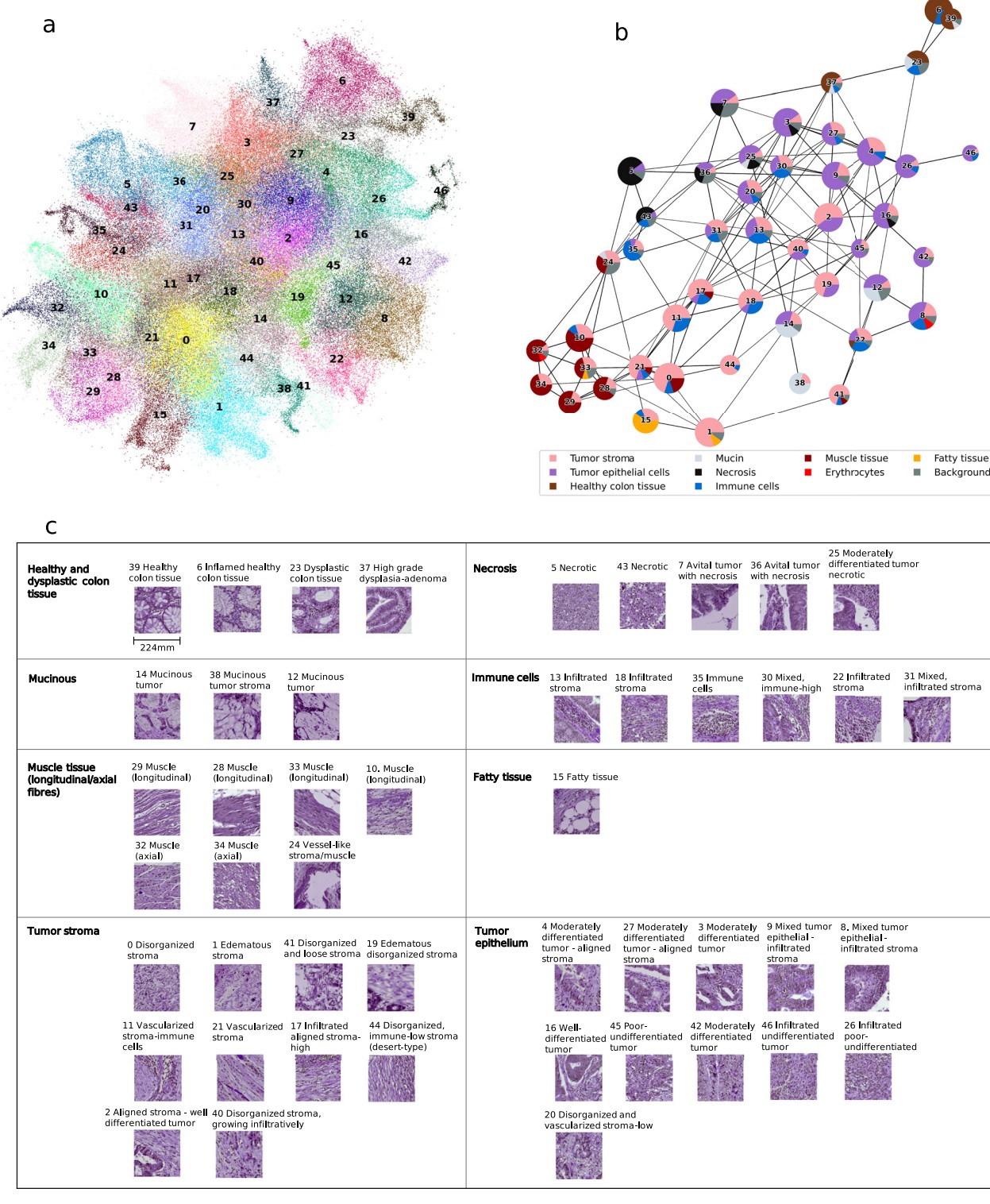

**Fig. 2 | Identification of HPCs in TCGA and subsequent classification into superclusters. a** UMAP showing 47 HPCs identified from the TCGA dataset, each scatter representing an image tile. **b** PAGA plot of HPCs. Each node represented an HPC with edges representing connections between HPCs based on their vector representation similarity. The pie chart of each node represented the tissue composition for each HPC. **c** Grouping of HPCs into super-clusters according to histopathology tissue similarities. Representative tiles for each HPC were labeled with ID and a brief description. The image tiles (224-by-224 pixels), at a magnification level of 10x (pixel size approximate 1.0 um), correspond to 224 mm in size (see scale bar in **c**). HPC, histomorphological phenotype cluster. PAGA, partition-based abstraction graph. TCGA, The Cancer Genome Atlas. UMAP, uniform manifold approximation and projection plot. Source data are provided as a Source Data file.

In total, we identified eight super-clusters: (1) healthy and dysplastic colon tissue, (2) necrosis, (3) mucinous areas, (4) immune cells, (5) muscle tissue (longitudinal/axial), (6) fatty tissue, (7) tumor stroma, and, (8) tumor epithelium, in no particular order, formed by groups of HPCs shown in Fig. 2c. Common histopathological characteristics were noted among HPCs within each designated supercluster, while HPCs encompassing varied tissue types across multiple super-clusters were often situated at their intersections. For instance, HPC 12, containing not only tumor epithelium but also mucinous tumor, was located between HPCs belonging to the two super-clusters of mucinous tumor and tumor epithelium in the PAGA plot. Moreover, HPC 23, marked by dysplastic colon tissue, formed a bridge between the healthy colon tissue HPC 39 and the tumor epithelium-containing super-cluster HPCs (e.g. HPCs 4, 26, 46), suggesting a potential chronological pathogenesis. In summary, the derived HPCs displayed distinctive histopathologic characteristics. Moreover, HPCs located in close proximity on the UMAP and PAGA plot, based on the extracted features, demonstrated common traits, hinting at potential pattern relationships, mixed phenotypes, or pathogenic trajectories.

## Assessment of HPC consistency within and across TCGA and AVANT cohorts

Although SSL methods have been applied recently in histopathology, there is usually no systematic analysis of the consistency of the histologic patterns discovered by these methods within and across datasets[21,27]. Here, we address this potential pitfall by incorporating several qualitative and quantitative assessments.

First, a qualitative assessment was conducted to evaluate the within-cluster and between-cluster heterogeneity of 47 HPCs derived in TCGA-COAD. Based on 32 randomly selected tiles from each HPC in the TCGA-COAD, three experts (ASLPC, JHJMvK, and MP) independently assessed each HPC by comparing tissue type quantity and architectures (histopathological assessment procedure stated above). Overall, all raters reached the general consensus that there was a noteworthy level of within cluster morphological similarity and a significant diversity among the 47 HPCs (Fig. 3 [a–i]), although phenotypic similarities varied across HPCs, implying that some HPCs may appear more similar than others. To delve deeper into the within-cluster and between-cluster heterogeneity, we carried out quantitative objective blinded tests within TCGA and AVANT tiles separately. This was to ascertain whether the morphological patterns identified by each HPC could also be recognized by human experts. In this test, the assessor (MP) was shown three groups of image tiles, each containing five tiles. Two groups were from the same HPC, and the third was from a randomly selected other HPC, also called the "odd HPC". The assessor was required to identify the "odd HPC" (Supplementary Fig. 2a, see Methods: Pathologist assessment of HPCs for details). Each of the 47 HPCs underwent 50 tests to determine the success rate. Within TCGA, we found that 17 out of the 47 HPCs achieved 100% identification rate, while the remaining 30 HPCs had a correct identification percentage ranging from 84% to 98%. Similarly, within in AVANT, 17 out of 47 HPCs achieved a perfect accuracy, while the rest had an accuracy ranging from 88% to 98% (Fig. 3j). In general, HPCs in close proximity to each other in the PAGA plot or belonging to the same super-cluster were more prone to erroneous assignment.

To evaluate the effectiveness of transferring morphological patterns from the TCGA to the external AVANT test set, three experts (ASLPC, JHJMvK, and MP) independently reviewed a randomly chosen set of 32 tiles from the TCGA-COAD subset and another 32 tiles randomly selected from the AVANT trial. This qualitative comparison concluded a remarkable resemblance between the TCGA and AVANT tiles within their respective HPCs (Fig. 3 [a–i]). In comparing the objective test results from TCGA and AVANT, we found an 80% overlap in the misclassified HPCs between the two datasets and 65% overlap in correctly classified HPCs (Fig. 3j, k). These results indicate that the

robust morphological features extracted from the training set can be effectively transferred to an independent unseen test set.

## HPC-based classifier was associated to OS in patients treated with standard-of-care and AVANT-experimental treatment

We explored the prognostic significance of HPCs on OS. The OS prediction model was developed within TCGA-COAD (see Methods: Optimization of HPCs and the prediction of OS using Cox regressions with L2 regularization for details). For external validation, we utilized the control group from the AVANT trial who had only received standard adjuvant chemotherapy (i.e. FOLFOX-4). The AVANT trial aimed to investigate whether combining bevacizumab, a humanized anti-vascular endothelial growth factor (VEGF) monoclonal antibody, with standard chemotherapy would improve survival among colon cancer patients[30,33]. The trial had three treatment arms: FOLFOX-4, bevacizumab+FOLFOX-4, bevacizumab+XELOX. The study was prematurely terminated due to the adverse effect in patient survival associated with bevacizumab[30]. Given the unique bevacizumab experimental treatment and its adverse effects, we hypothesized that the survival model trained on TCGA-COAD patients may not generalize well in the AVANT bevacizumab-treated group. We referred to our source population, represented by the TCGA-COAD, as the "standard-of-care group", in contrast to the unique bevacizumab treatment in the AVANT trial. We then opted to validate the OS prediction model, trained on TCGA-COAD data, primarily on AVANT patients who exclusively received standard FOLFOX-4 chemotherapy. This subset is henceforth referred to as the "AVANT control group", serving as an independent test set for validation.

We modeled HPCs on OS using Cox regression with L2 regularization trained within the TCGA-COAD incorporated all 47 HPCs as predictors. The model was optimized through five-fold cross-validation (CV) on the TCGA training set (Supplementary Fig. 3a). The optimized regularized Cox model was then tested in the independent test set of the AVANT control group. We observed a test set c-index of 0.65 (bootstrap 95% confidence interval [CI] = 0.55–0.74) (Supplementary Fig. 3b). The HPC-based classifier (i.e. high risk versus low risk) was determined by the median predicted hazard obtained in the TCGA-COAD. The HPC-based model also outperformed a clinical baseline model trained on age, sex, tumor-stroma ratio (TSR), and AJCC TNM stage (c-index = 0.58, bootstrap 95% confidence interval [CI] = 0.49–0.67) (Supplementary Fig. 3c). To investigate whether our HPC-based classifier provides additional prognostic value to the existing important clinical predictors, a regular multivariable Cox regression was fitted within the external AVANT control test set. The model included HPC-based classifier as well as important clinical and demographic variables (Fig. 4a). Notably, the HPC-based risk classifier demonstrated significance as an independent prognostic factor (hazard ratio [HR] = 2.50, 95% CI = 1.18–5.31), along with male sex (HR = 2.42, 95% CI = 1.07–5.47) and the TSR (HR = 2.49, 95% CI = 1.23–5.04). The 20 most important HPCs associated to OS were summarized using the interpretable SHapley Additive exPlanations (SHAP) (Fig. 4c).

Harnessing the well-defined experimental protocols in AVANT, we were granted the unique opportunity to examine the influence of HPCs on colon cancer OS in the bevacizumab treatment groups (i.e. bevacizumab+FOLFOX-4 or bevacizumab+XELOX). Given the proven comparable therapeutical efficacy of FOLFOX-4 and XELOX[33–35], we consolidated the bevacizumab+FOLFOX-4 and bevacizumab+XELOX cohorts into a unified "AVANT-experimental group". Similar to the analysis stated above, we trained Cox regressions with L2 regularization encompassing all 47 HPCs within the AVANT-experimental patients using 5-fold CV (Supplementary Fig. 3d). The HPC-based classifier remained of independent prognostic value (HR = 1.82, 95% CI = 1.11–2.99) after adjusting for age, sex, TNM tumor staging, and TSR (Fig. 4b). The top 20 most influential HPCs on OS prediction were shown in the SHAP summary plot (Fig. 4d).

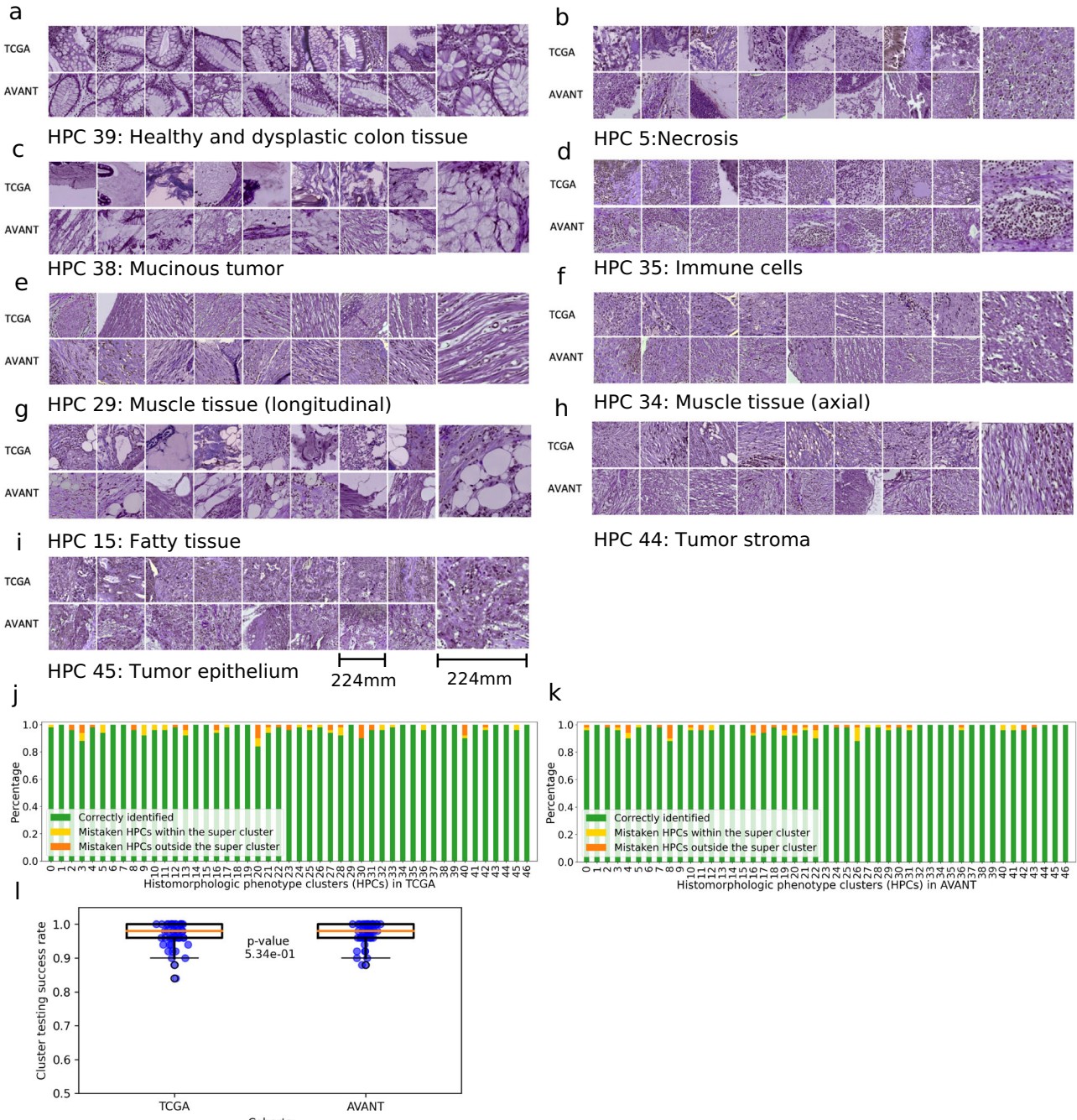

**Fig. 3 | Verification of HPCs in the TCGA training set and the external clinical AVANT trial. a–i** Example tiles from TCGA (upper row) and AVANT (lower row) showcase the eight super-clusters with a zoomed-in representative tile. The image tiles (224-by-224 pixels), at a magnification level of 10x (pixel size approximate 1.0 um), correspond to 224 mm in size (see scale bar in **a–i**). The muscle tissue super-cluster is further divided into longitudinal and axial subgroups. **j, k** Stacked bar plots illustrate instances of misclassification for each HPC in TCGA training set and AVANT external test set. Green bars represent the percentage of correctly identified odd clusters, yellow bars indicate misclassifications within the tested HPC's super-cluster, orange bars show misclassifications outside the super-cluster. **l** Box plots display similar distributions of success test rates (corresponding to the green bars in panels j and k) for HPCs in TCGA and AVANT cohorts (two-sided Wilcoxon signed-rank test, *p* = 0.534). Each blue point within each box plot represents the success test rate for a single HPC, calculated based on 50 tests per HPC. The central orange line within each box represents the median, while the bounds of the box indicate the 25th and 75th percentiles (interquartile range). Whiskers extend to the minima and maxima within 1.5 times the interquartile range. Points outside the whiskers represent outliers. HPC, histomorphological phenotype cluster. TCGA, The Cancer Genome Atlas. AVANT, Bevacizumab-Avastin® adjuVANT trial. Source data are provided as a Source Data file.

## Pathological assessment of OS-associated HPCs in the standard-of-care and AVANT-experimental groups

We highlighted the top 20 most influential HPCs on OS prediction reflecting the SHAP summary plot (Fig. 4c, d) in the PAGA plots for the standard-of-care and AVANT-experimental groups (Fig. 5). The survival-favorable HPCs were highlighted in a shade of blue and the survival-unfavorable HPCs were highlighted in red. Preliminary findings showed that HPCs containing proportionally more healthy colon tissue or immune cells appeared associated with an improved OS, and HPCs comprising mucinous tumor, tumor stroma, and poor-to-undifferentiated (high-grade) tumor epithelium were associated with worse OS in both groups. An overview of the HPCs and their

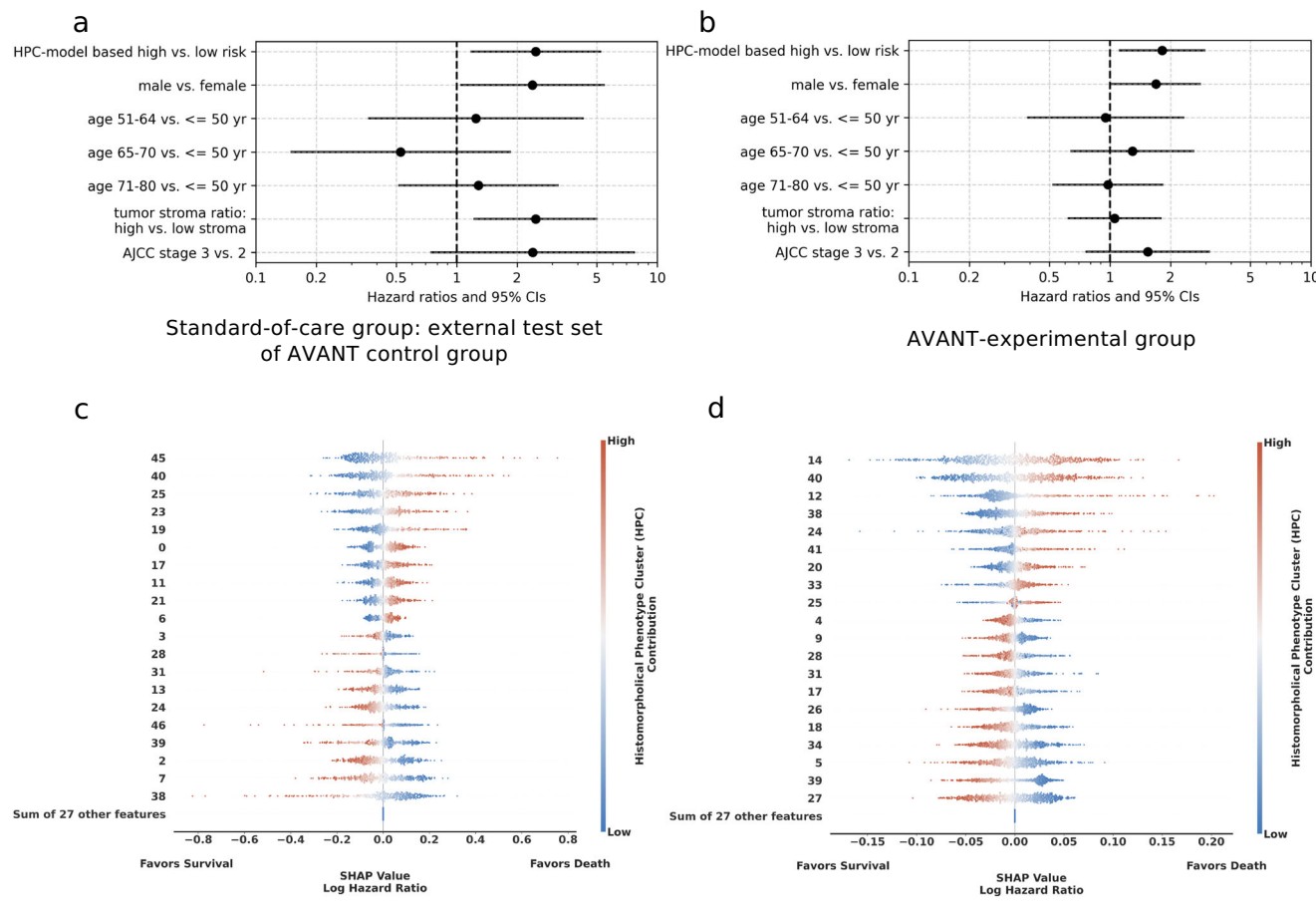

**Fig. 4 | HPC-based classifier was associated with OS in patients treated with standard-of-care and AVANT-experimental treatment. a** Ordinary Cox regression for OS, incorporating the HPC-based risk classifier, along with sex, age categories, tumor-stroma ratio, and AJCC TNM staging, was conducted within the external test set of the AVANT control group (*N* = 379 patients after excluding those with missing clinical information). Each point represents the point estimate of HR, and the horizontal whiskers depict the 95% CI. The HPC model-based classifier stands as an independent prognostic factor (HR = 2.50, 95% CI = 1.18–5.31) for OS. **b** Ordinary Cox regression for OS, incorporating the HPC-based risk classifier, along with sex, age categories, tumor-stroma ratio, and AJCC TNM staging, was conducted within the AVANT experimental group (*N* = 751 patients after excluding those with missing clinical information). Each point represents the point estimate of HR, and the horizontal whiskers depict the 95% CI. The HPC model-based classifier stands as an independent prognostic factor (HR = 1.82, 95% CI = 1.11–2.99)

for OS. **c** and **d** The SHAP summary plots depict the relationship between the center-log-transformed compositional value of an HPC and its impact on death hazard prediction. Statistics estimated from AVANT control (*N* = 405 after excluding those with missing survival data) (**c**) and experimental groups (*N* = 780 patients after excluding those with missing survival data) (**d**). The color bar indicates the relative compositional value of an HPC, with red indicating higher and blue indicating lower composition. Higher compositions of the top 10 HPCs were associated with worse OS, while higher compositions of the bottom 10 HPCs were linked to improved OS. AJCC TNM, American Joint Committee on Cancer tumor-node-metastasis classification. AVANT, Bevacizumab-Avastin® adjuVANT trial. HPC, histomorphological phenotype cluster. OS, overall survival. SHAP, SHapley Additive exPlanations. TCGA, The Cancer Genome Atlas. Source data are provided as a Source Data file.

associations to OS, including pathological explanations and references, were given in Supplementary Table 2. We will highlight some OS-associated HPCs in depth here.

The healthy colon tissue-containing HPC 39 was among the top survival-favorable HPCs as indicated by larger SHAP values in both groups (Fig. 4c, d). In the standard-of-care group, HPC 23 characterized by dysplastic and low-grade tumor epithelium and HPC 6 marked by stroma-infiltrated healthy colon tissue indicating inflammation, were associated with a worse OS. HPC 31, predominantly composed of immune cells, was associated with an improved OS in both treatment groups. A similar association was also noted for the infiltrated-stroma HPC 13, which is part of the immune super-cluster, in the standard-of-care group (Fig. 5a) as well as HPC 18 (infiltrated stroma, containing aligned tumor stroma and immune cells) and HPC 17 (infiltrated aligned stroma-high; organized tumor stroma with an immune component) in

the AVANT-experimental group (Fig. 5b). In the AVANT-experimental group, the mucinous super-cluster (HPCs 12, 14 and 38) was associated with a poor survival (Fig. 5b), while conversely, HPC 38 (mucinous tumor stroma) led to a better survival in the standard-of-care group (Fig. 5a).

Upon closer examination, we noted that the algorithm captured a distinction in stroma organization within the stromal tissue presented in the tumor stroma and tumor epithelium super-clusters. HPCs in these two super-clusters all contained some component of stromal tissue. However, HPCs with disorganized or heterogeneous tumor stroma with neovascularization (HPC 40 in both groups; 0, 11 and 21 in standard-of-care, and 41 in the AVANT-experimental group) were associated with a poor survival, whereas aligned and organized tumor stromal "strands" were often observed among the top survival-favorable HPCs (HPC 2 in the standard-of-care and HPC 27 in the AVANT-experimental group).

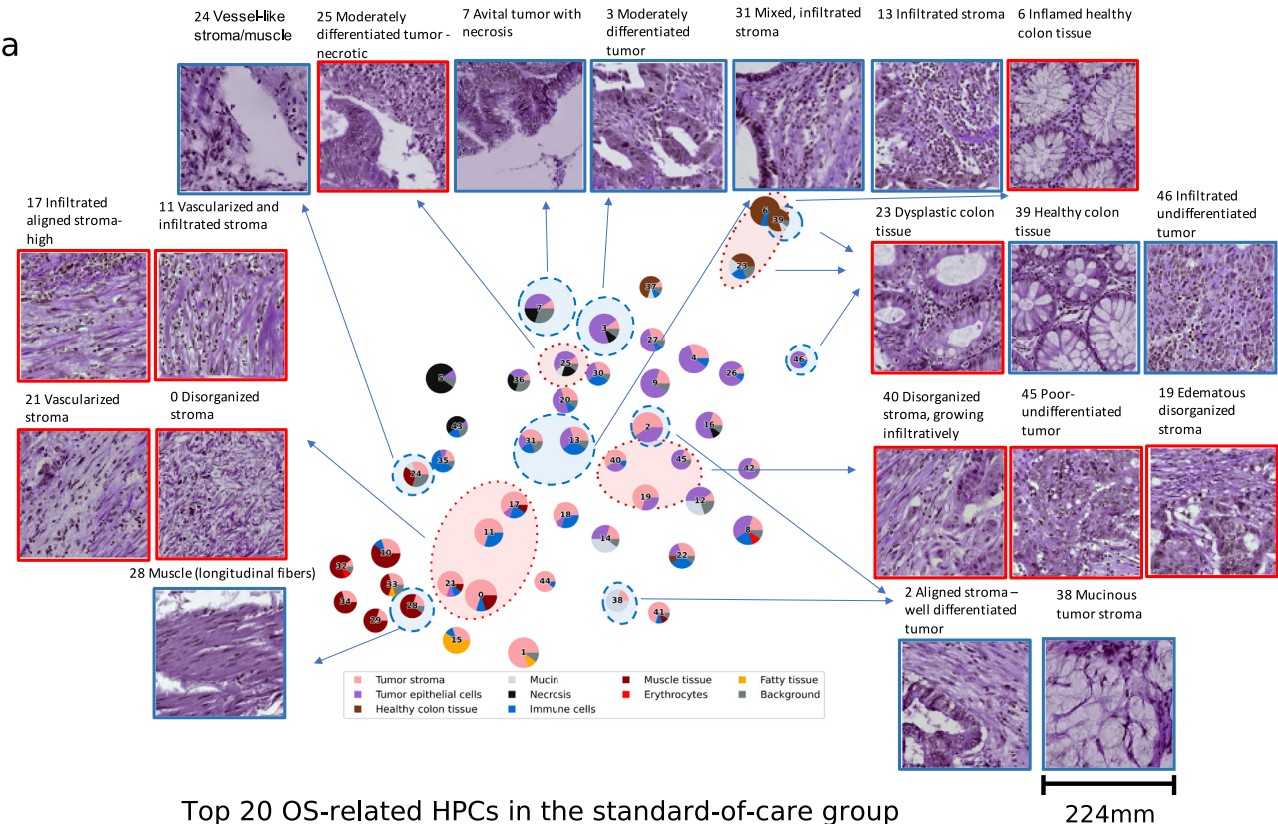

Top 20 OS-related HPCs in the standard-of-care group

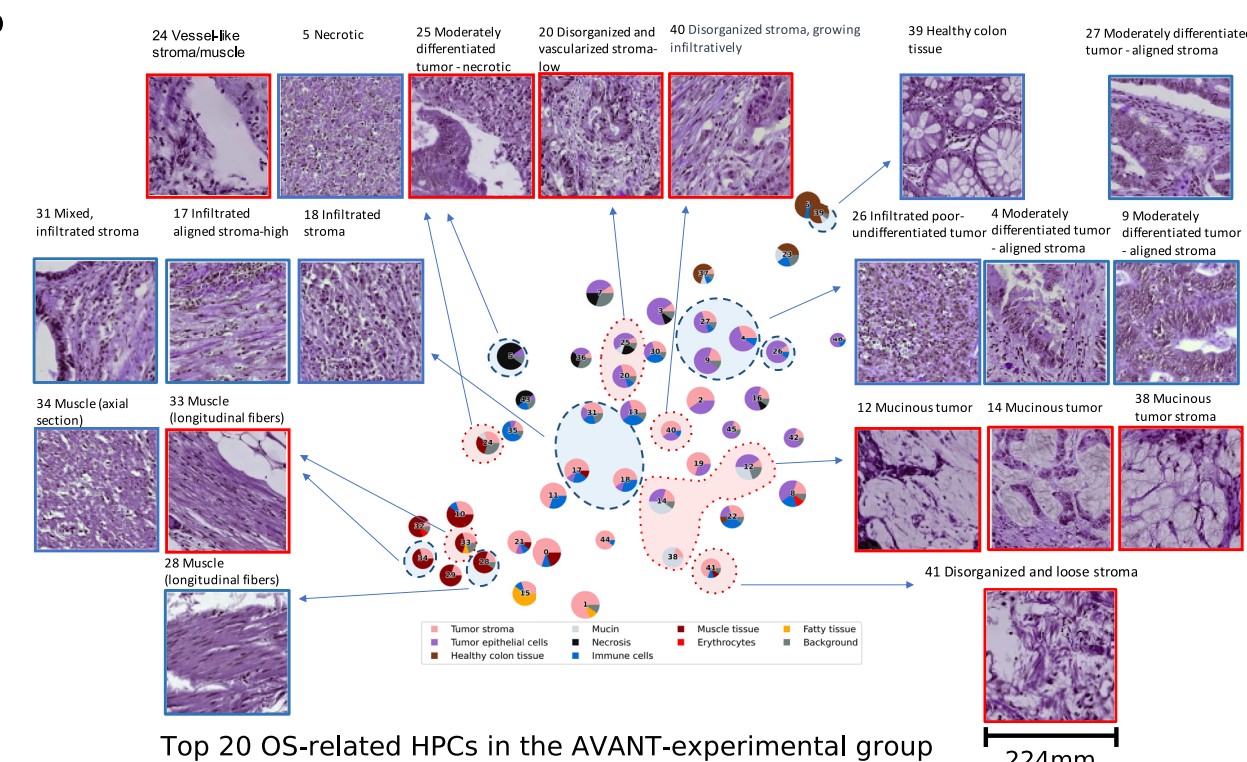

Top 20 OS-related HPCs in the AVANT-experimental group

**Fig. 5 | PAGA plots highlighted with important HPCs related to OS in the standard-of-care and experimental treated group. a** Standard treated group: HPCs colored in the red are linked to worse survival and HPCs colored in blue are linked to better survival. **b** AVANT experimental treated group: HPCs colored in the red are linked to worse survival and HPCs colored in blue are linked to better

survival. The image tiles (224-by-224 pixels), at a magnification level of 10x (pixel size approximate 1.0 um), correspond to 224 mm in size (see scale bar in **a** and **b**). AVANT, Bevacizumab-Avastin® adjuVANT trial. HPC, histomorphological phenotype cluster. PAGA, partition-based graph abstraction. Source data are provided as a Source Data file.

Moreover, the tumor differentiation grade was another parameter correlating with HPCs and their impact on OS. Poor-to-undifferentiated (high-grade) tumor epithelium, e.g. HPC 45, in the standard-of-care group, was linked to a worse survival, while well-to-moderately differentiated (low-grade) tumor led to an improved survival (e.g. HPCs 3 in the standard-of-care group and HPC 4 in the AVANT-experimental group). A unique pattern was furthermore observed in survival-favorable HPCs 46 (standard-of-care group) and 26 (AVANT-experimental group), which was characterized by a predominance of poor-to-undifferentiated tumor epithelium but accompanied by a notable influx of immune cells.

Additional findings were related to the survival-favorable HPC 7 (avital tumor with necrosis) in the standard-of-care group (Fig. 5a) and the positive association between necrosis-dominated HPC 5 and better survival in the AVANT-experimental group (Fig. 5b). The associations between HPCs containing muscle tissue (e.g. HPC 24, 33 and 34) and survival were generally inconsistent, possibly due to the similarity of muscle fibers and tumor stroma, caused for instance by the organization of muscle fibers and vascularization.

### Outcome-associated HPCs are linked to diverse immune features in the tumor microenvironment

Spearman correlation coefficients were calculated between the top 20 survival-related HPCs on one hand, and the TCGA immune landscape on the other hand (see Methods: Linkage between survival-related HPCs, immune landscape, and gene expression data for details). The correlation heatmaps were plotted with bi-directional hierarchical clustering (Fig. 6a, b). Interestingly, in the standard-of-care group (Fig. 6a), HPCs 0, 11, 17 and 21, all part of the phenotypic tumor stroma super-cluster, were identified by the genotypic immune analysis as having a positive correlation with the stroma-high category. Moreover, their SHAP values indicated a positive association with worse survival. Conversely, HPCs 13 and 31 in the immune cell super-cluster were correlated with higher leukocyte fraction. Survival prediction model suggested the immune cell super-cluster was in general associated with a better survival. Further anticipated findings included the validation of HPCs 2 and 24, marked by stroma-high tiles, through immune feature correlation within the stroma-high category.

In the AVANT-experimental group (Fig. 6b), survival-favorable HPCs 5, 17, 31, 18 correlated with an increased leukocyte fraction. Notably, HPCs 17 and 31 aligned with an elevated immune cell composition (Fig. 5b). Additionally, HPCs 17 and 18 were also associated with a higher expression of stroma, which was consistent with the observed stroma-high morphology (Fig. 5b). On the contrary, survival-unfavorable HPCs 14, 40 and 20 exhibited higher genomic instability. For instance, HPCs 40 and 20 were positively correlated with homologous recombination defects, intratumor heterogeneity, and HPC 14 was positively linked to nonsilent mutation rate, single nucleotide variants, and indel neoantigens. Although HPC 14 also showed a positive correlation with the leukocyte fraction, a more pronounced association with genomic instability through multiple pathways seemed to play a greater role in its negative impact on OS.

The resulting correlations with immune landscape data aligned with the observed morphologies of HPCs, particularly in stromal and immune features. Furthermore, the data suggest a potential role in genomic instabilities within the AVANT-experimental group. Taken together, these results demonstrate that HPCs can capture the remarkable heterogeneity of the tumor microenvironment.

### Outcome-associated HPCs are linked to oncogenic pathways and bevacizumab's mechanism of action

Next, we performed gene set enrichment analysis (GSEA) to discern associations between top OS-related HPCs from both the standard-of-care and AVANT-experimental groups and key cancer hallmark pathways (Fig. 6c, d). In the AVANT-experimental cohort, survival-related

HPCs showed striking alignment with the enrichment observed in oncogenic hallmark pathways (Fig. 6d), which such alignment was overall much less pronounced in standard-of-care group (Fig. 6c). Still, in the standard-of-care group, pathways encoding epithelial-to-mesenchymal transition, leading to an increased tumor-stromal percentage, were enriched in survival-unfavorable HPCs 11, 17, 40 and 19. HPCs related to inflammatory response pathways showed primarily positive associations with OS, e.g. with HPCs 13 and 31 related to better survival.

In the AVANT-experimental group (Fig. 6d), we observed a strong correlation between hierarchical clustering based on oncogenic enrichment scores and survival-related HPCs. Several HPCs exhibited enrichment in pathways that potentially be specific to bevacizumab through its target of $VEGFa$ expression and $KRAS$ signaling-up pathway. HPC 5, characterized by necrosis, was linked to elevated $VEGFa$ expression (rho=0.163, two-sided $p = 0.005$), indicating that patients with higher pre-treatment necrosis levels may have benefited from the bevacizumab treatment. HPC 27, 18, and 31 were associated with enriched hypoxia (HPCs 31, 27) and angiogenesis (HPCs 18, 31), and unfolded protein (HPCs 31, 1, 5) pathways which involve the $VEGFa$ gene. In addition, other survival-favorable HPCs may not be specific to bevacizumab but related to the standard cytotoxic chemotherapy of XELOX and FOLFOX-4. Survival-unfavorable HPCs (12, 38, 14, 24, 20) were linked to the $KRAS$ signaling-up pathway, which may have a negative impact on the prognosis of patients treated with FOLFOX. Certain survival-favorable HPCs exhibited enrichment in pathways related to cell cycle regulation, signaling, DNA repair, and growth, including G2/M checkpoint (34, 31, 5), E2F targets (5, 31, 34, 26, 18), Myc targets (5, 31, 34, 26, 18), and mTORC1 signaling (5, 31, 34, 26, 18), DNA repair (18, 31). In contrast, survival-unfavorable HPCs demonstrated depletion in these pathways.

## Discussion

In this study, we derived and independently validated a total of 47 distinct HPCs that were extracted from colon cancer H&E WSIs using a self-supervised algorithm. These HPCs possess distinctive histomorphologic features carefully identified and assessed by the pathologists and were also linked several immune features and oncogenic pathways. The HPCs showed state-of-the-art performance on OS prediction. Furthermore, the HPC-based risk classifier was an independent prognostic factor after adjusting for important clinical and demographic variables, suggesting additional insight beyond the current clinical prognostication. The unique AVANT trial enabled us to endeavor in identifying possible mechanisms of response to bevacizumab and standard chemotherapy using the HPCs.

Moreover, we hereby emphasize the importance of the tumor microenvironment, or tumor stroma, and its effect on survival. Tumor stroma is composed of extracellular matrix, vasculature, immune cells and cancer-associated fibroblasts, forming a complex and close interaction with tumor epithelial cells[36,37]. Subject to increasing research the past decades, this dynamic entity has been found to modulate tumor behavior through its cross-talk, and ultimately influence patient-related outcomes. Specifically, regarding the amount of stroma and stromal architecture, i.e. alignment or categorization of the desmoplastic reaction, and immune infiltrate in the stroma, our results corroborate previous literature[31,38–47]. However, such assessments have not been implemented in standard pathology diagnostics.

We identified common histopathological patterns associated with survival as observed in both the standard and bevacizumab-treated group. In line with our results, immune cell infiltration (e.g. HPCs 13, 31)[40,46] and aligned, stroma-low tumors (e.g. HPCs 2, 27)[43,47] were associated with better survival, while poor-to-undifferentiated (high-grade) tumor epithelium (e.g. HPC 45) or mucinous tumor epithelium (e.g. HPCs 12, 14)[1,3,48,49], and disorganized stroma-high tumors (e.g. HPCs 40, 0)[43,47] were linked to worse survival. This

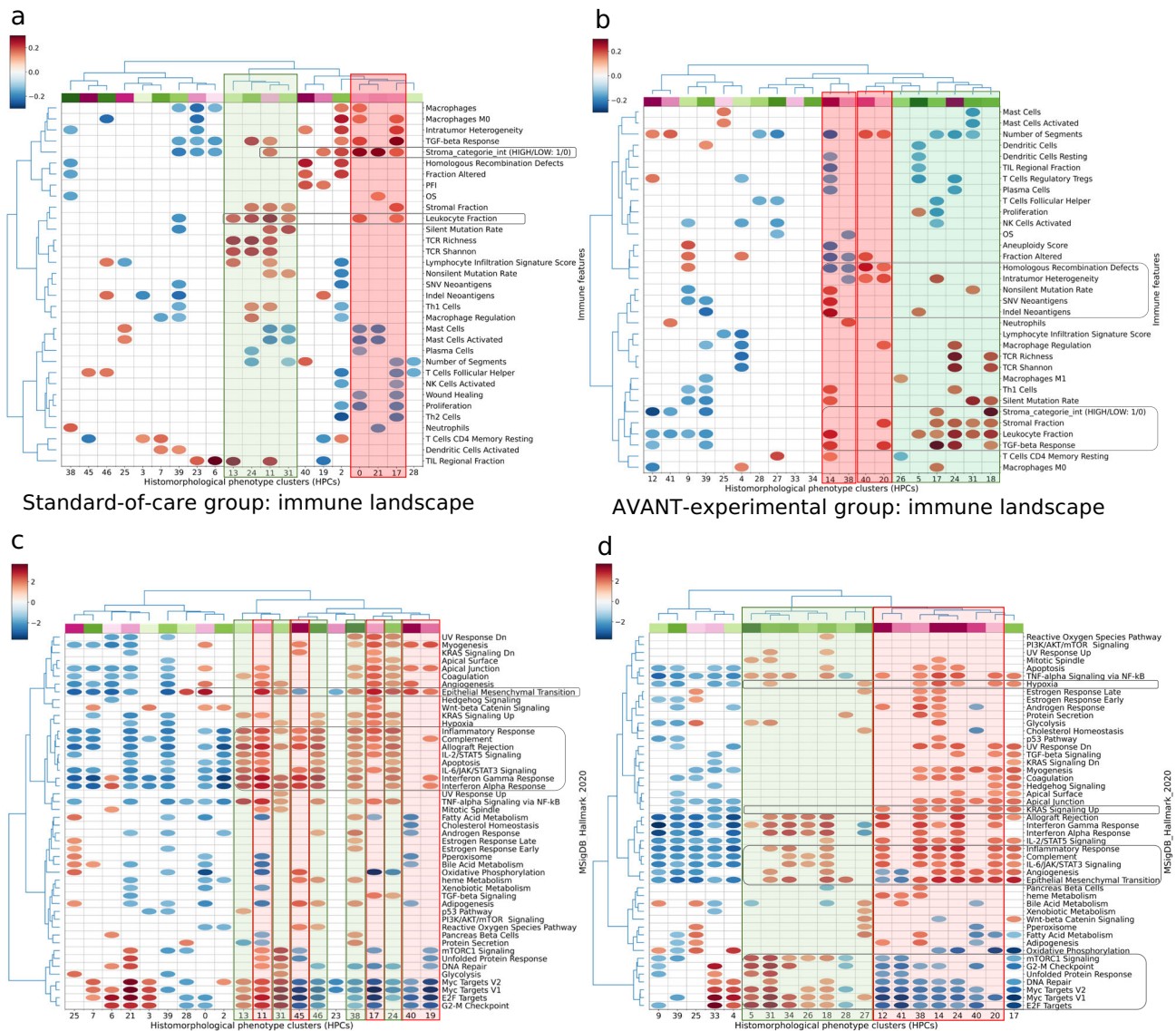

**Fig. 6 | Survival-associated HPCs in relation to immune and genetic profile.** **a** Standard-of-care group: Spearman's correlations between top 20 OS-related HPCs and immune landscape features. HPCs (columns of the matrix) were colored according to the beta-coefficients estimated from the optimized regularised Cox regression, with red indicating HPCs related to worse survival and green indicating HPCs related to better survival. The color bar at the upper left corner indicates the value of correlation coefficients with red denoting positive and blue denoting negative correlations. **b** AVANT-experimental treated group: Spearman's correlations between top 20 OS-related HPCs and immune landscape features. **c** Standard-of-care group GSEA between the top OS-related HPCs and major cancer hallmark pathways. HPCs (columns of the matrix) were colored according to the beta

coefficients estimated from the optimized regularized Cox regression, with red indicating HPCs related to worse survival and green indicating HPCs related to better survival. The color bar at the upper left corner indicates the value of the correlation coefficients with red denoting enrichment and blue denoting under-representation in a gene pathway. **d** AVANT-experimental treated group GSEA for the top 20 OS-related HPCs. The immune landscape analysis ($N = 355$ patients) and GSEA analysis ($N = 265$ patients) were performed using data available from TCGA. AVANT, Bevacizumab-Avastin® adjuVANT trial. GSEA, gene set enrichment analysis. HPC, histomorphological phenotype cluster. OS, overall survival. Source data are provided as a Source Data file.

pathological phenotype correlated with the corresponding genetic immune profile (e.g. increased leukocyte fractions correlated with HPCs 13 and 31; and a stroma-high category was seen in HPCs 0, 21, 17, 11) and enrichment in oncogenic pathways (e.g. epithelial-to-mesenchymal transition pathway, contributing to tumor stroma amounts[36], correlated with HPCs 17 and 11). We also observed an association, though imperfect, between mucinous tumor, poor-to-undifferentiated tumor epithelium, and survival. One explanation may arise from the absence of contextual information in small images. The differentiation between well-differentiated and undifferentiated, or adenocarcinoma and more mucinous tumor types, is

established based on whether each tissue type constitutes more than 50% of the total tumor, underscoring the importance of considering the overall context[1,3,48,49].

Another interesting discovery emerges from the survival-favoring HPCs 26 (AVANT-experimental group) and 46 (standard-of-care group). These HPCs contain primarily poor-to-undifferentiated tumor epithelium but with a high influx of immune cells. Such a histopathological pattern is frequently observed in MSI tumors[50]. MSI-high tumors have been linked to a favorable prognosis[2,51], characterized by lower differentiation grade, increased T-cell infiltration, and reduced susceptibility to invasiveness and *KRAS* mutation[51], however, are

commonly identified through separate MSI analysis and/or additional immunohistochemical staining for mismatch repair enzymes in pathology diagnostics[2,3,51].

Furthermore, we also noted HPC 39 containing predominately healthy colon tissue associated with better survival. Interpretation hereof lies in the nature of the multivariable analysis, where the 47 HPCs were modeled simultaneously. One can interpret this result as, while holding the other 46 HPCs constant, patients with relatively more healthy colon tissue, showed an improved survival. The higher proportion of healthy colon tissue may thus indicate relatively smaller or less aggressive tumors. Indeed, within routine TNM assessments, lower pathological T-stage is known to lead to an improved survival[1,3]. Additionally, the favorable OS could potentially also be linked to an absence of the cancer field effect, leaving this healthy colon tissue unaffected from the cancerous lesion[52].

In the GSEA analysis, we noted a remarkable concurrence in AVANT-experimental group between clustering based on cancer hallmark pathways and outcome-related HPCs, while such alignment was much less pronounced in the standard-of-care group. One possible explanation is the heterogeneity of patients in the TCGA-COAD dataset. In contrast to the well-defined treatment protocol in AVANT, TCGA-COAD patients encompass diverse disease and demographic profiles. Consequently, this diversity led to a wide spectrum of treatments, including surgical, neoadjuvant, and adjuvant therapies. The survival-related HPCs discovered in the TCGA-COAD could therefore be, if at all, related to multiple distinctive biological pathways. A general alignment observed in the AVANT-experimental group was hence not anticipated in the standard-of-care group.

In the AVANT-experimental group, however, several survival-favoring HPCs either directly correlated with *VEGFa* expression or were associated to enrichment in oncogenic pathways involving *VEGFa* gene, indicating a favorable responses to the contentious bevacizumab treatment. In particular, HPC 5, primarily characterized by necrosis, emerged as a significant contributor to enhanced survival, displaying a positive correlation with *VEGFa* expression, which is the target of bevacizumab. Necrosis promotes the expression of *VEGFa*, as dying tumor cells release signals that stimulate the growth of new blood vessels[33,53]. In addition, survival-favoring HPCs 18, 27 and 31 were associated with enriched hypoxia and angiogenesis pathways, which also involve the *VEGFa* gene. We hypothesize that patients exhibiting a higher abundance of HPC 5, 18, 27, and 31 may correspondingly express elevated levels of bevacizumab's target *VEGFa*, which in turn results in a more favorable response to this treatment.

Oxaliplatin plus 5-fluorouracil-based regimens of XELOX and FOLFOX-4 are standard chemotherapy for colorectal cancer[2]. Nonetheless, the treatment response is still modest with an estimate rate of approximately 50%[1,54], and the prediction of which patients will respond to this adjuvant chemotherapy remains challenging. We observed enrichment in oncogenic pathways that may be within the context of XELOX and FOLFOX-4 treatments in the AVANT trial. Survival-unfavorable HPCs (12, 38, 14, 24, 20) were linked to the *KRAS* signaling-up pathway. In line with previous literature, several *KRAS* mutations activate downstream signaling pathways and the *KRAS G12D* mutation was predictive of an inferior response to FOLFOX[55]. HPCs 12, 14, and 38 contained mucinous tumors tissue, which has also been linked to *KRAS* mutational burden[50,56]. Survival-favoring HPCs (5, 31, 34, 26, 18, 27) were associated with enrichment in pathways of cell cycle, signaling, DNA repair, and growth (i.e. G2/M checkpoint, E2F targets, Myc targets, and mTORC1 signaling), while survival-unfavorable HPCs (12, 41, 38, 14, 24, 40, 20) linked to a depletion of those pathways. Interestingly, patients with altered DNA repair capacity showed greater benefits from treatment with oxaliplatin[54,57]. Moreover, carriers of *MNAT1* gene, which is one of the leading genes in the G2/M checkpoint pathway[58], were linked to better treatment outcome of FOLFOX[54]. A plausible mechanism is that patients exhibiting activated oncogenic activities within these pathways might harbor a greater abundance of targets suitable for oxaliplatin-based cytotoxic chemotherapy.

Another finding was our HPC-based prediction on OS outperforming the clinical baseline model. This HPC-based risk classifier remained an independent prognostic factor (HR = 2.50, 95% CI = 1.18–5.31) after adjusting for crucial clinical and demographic variables including TSR and tumor stage. This finding aligns with recent findings reported by Jiang et al.[13]. Of interest, although both HPCs and the TSR scores were derived from H&E slides, they appeared to encapsulate distinct non-overlapping information. A potential explanation could be that the HPC-based classifier captures intricate details, while TSR assessment requires a broader contextual understanding, not fully attainable with small image patches[37,38,59]. Comparing our OS prediction directly with previous studies poses further challenges due to differences in cancer types (e.g., colorectal[12,13,60] instead of colon cancer only), varied outcomes (e.g., 5-year disease-free survival [DFS][12]), diverse statistical measures (e.g., hazard ratios[60]), and absence of independent test sets[15] in prior studies. Nevertheless, a recently study reported a test set c-index of 0.65 for OS prediction in colorectal cancer[13] aligning with our reported c-index in the context of colon cancer.

Building upon the aforementioned findings, this study showcases the prospective clinical utility of AI-generated HPCs (Fig. 7). Cancer WSIs were preprocessed into image patches and subsequently used to train SSL encoders and to form HPCs. These HPCs serve as condensed representations of the original WSIs, ready to be inspected by pathologists and enabling flexible linkage to multimodal omics data. These HPCs hold promise in classifying various tumor characteristics, potentially predicting patient prognosis and discerning distinct sensitivity groups to various therapies. Although this study was already trained on multicenter TCGA data and validated in an external, clinical multicenter cohort, we plan to conduct additional validations in other population-based external cohorts to strengthen its clinical applicability. In a pathology lab, this could be implemented as followed: alongside the routine pathology report, an AI-generated report would provide personalized prognostic risk quantification (e.g., based on the patient's HPC composition and SHAP values). The report would also include tissue composition descriptions for each HPC, granting pathologists a complete overview per patient. Subsequently, with these reports used in the multidisciplinary team meetings, a colon cancer patient can be granted an optimal personalized treatment strategy.

Given the significant potential implications of these HPCs on OS and therapy response, implementation of this AI-based analysis may be advocated to international guideline organizations, such as the TNM evaluation committee of the Union for International Cancer Control (UICC). Our HPC-based analysis not only summarizes the TNM classification, but also correlates to parameters assessed in standard pathology diagnostics, such as the International Collaboration on Cancer Reporting (ICCR)[61]. Of note, ICCR parameters pertain to a global assessment on whole slide or patient level, while HPCs can provide a local assessment of histopathological characteristics on the tile level. However, these tile-level histopathological characteristics can still be linked to ICCR parameters. For example, a core element of the ICCR, 'Histological tumor grade', can be correlated to HPCs containing tumor epithelium with different differentiation statuses. HPCs 2 and 3 are for instance characterized by well-moderately differentiated (or 'low-grade') tumor epithelium, while HPC 45 is formed by poorly-to-undifferentiated (or 'high-grade') tumor epithelium-containing tiles. Moreover, 'Histological tumor type' in the ICCR can be linked to HPC 12, which contains predominantly mucinous tumors. Lastly, the TNM classification, and particularly tumor size, in the ICCR reporting guidelines may be linked to HPCs containing healthy colon tissue (e.g. HPC 39), since WSIs with more of the healthy HPC 39 may correspond to a fractionally smaller tumor size (i.e. lower T-stage).

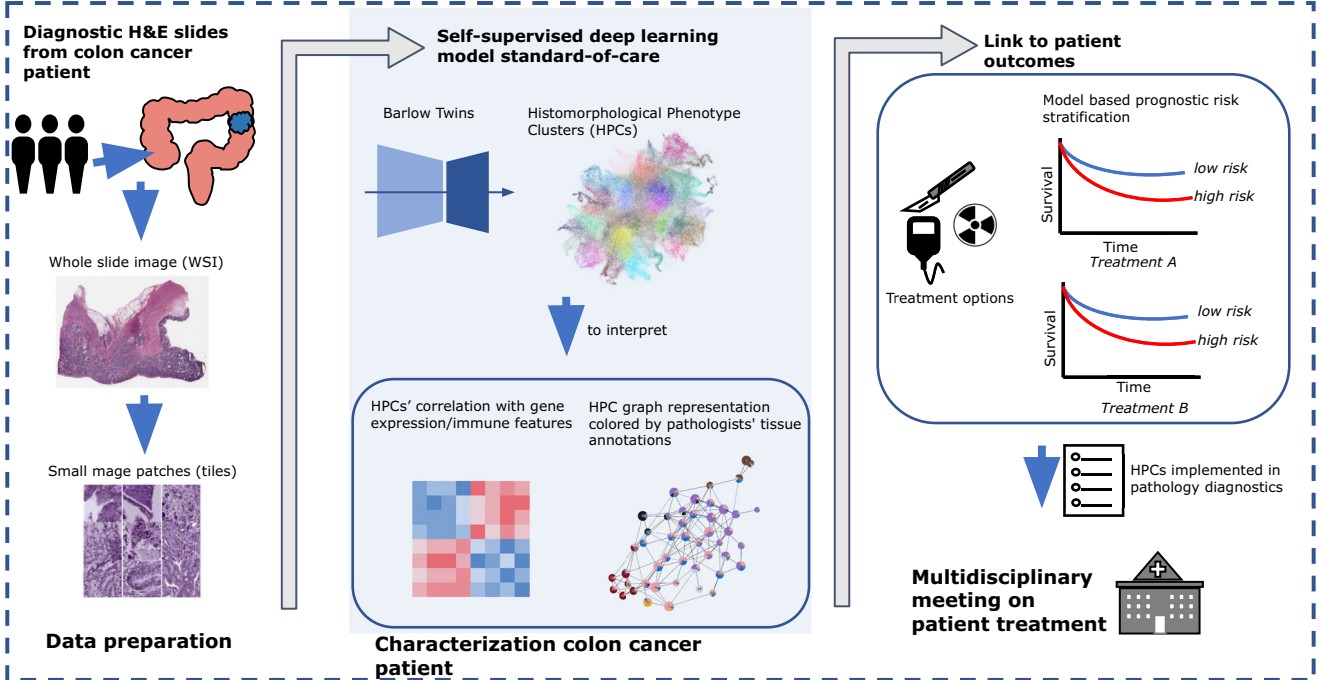

**Fig. 7 | Clinical application of AI-derived HPCs in prediction of patient outcomes.** The clinical algorithm consists of three key stages: data preparation, cancer patient characterization, and AI-supported multidisciplinary treatment meetings. Data preparation involves collecting histopathology WSIs, segmenting them into small image tiles. Patient characterization encompasses SSL model training, yielding HPCs via clustering. HPCs are easily interpretable by pathologists, linkable to omic data. Most importantly, HPCs are valuable for predicting diagnosis, patient outcomes, and treatment responses. In treatment-related outcomes, AI-predicted high/low risk groups aid multidisciplinary meetings, enabling personalized treatment plans by oncologists, pathologists, and other physicians. AI, artificial intelligence. HPC, histomorphological phenotype cluster. SSL, self-supervised learning. WSI, whole slide image.

Despite the interesting findings, the study also has several limitations. The identification of HPCs was based on small image tiles as is imperial to model training, while information regarding the larger context is likely lost, as also stated above. For example, it is often challenging to distinguish aligned and organized tumor stromal strands from muscle tissue through traditional microscopic assessment[59]. Therefore, pathologists typically make this distinction based on contextual cues, color variances, or use additional immunohistochemical stainings, all of which is not available within the small image patches. Another limitation pertains to using TCGA as the training set. Although the TCGA is a large opensourced database, it depends on the availability of registered clinicopathological data. This introduces potential bias and variability in data quality across participating institutions. Moreover, due to data availability, we were only able to focus on predicting OS rather than DFS which may better reflect tumor behavior and biology[29,62]. Lastly, due to the AVANT treatment regimen design, separate analysis regarding bevacizumab or oxaliplatin-based chemotherapy was not attainable. Nonetheless, we made efforts to differentiate between their distinct mechanisms by conducting histopathological inspections and correlating the findings with immune landscape and oncogenic pathways.

In conclusion, our study employed a self-supervised approach to identify and validate histopathological features in colon cancer that are recognizable by human eyes and relevant to prognosis. These features were interpreted through a pathology-focused perspective. Our results highlighted the clinical significance of tumor tissue type, stromal amount and architecture, and the involvement of immune cells. Integration of histopathological features with genetic and gene expression data unveiled potential insights into oncogenic pathways and their relation to patient survival. Utilizing data from the clinical AVANT trial, we proposed mechanisms influencing patient sensitivity to diverse treatments. Future research should focus on refining prediction accuracy and validating the proposed mechanisms regarding the therapeutic strategies in colon cancer.

## Methods

### Ethics statement

Our research complies with all relevant ethical regulations. The study protocol was approved by Applied Bioinformatics Laboratories of New York University Grossman School of Medicine and the Department of Surgery of Leiden University Medical Center. The analyses were performed using anonymized archival material, not necessitating additional informed consents. Data from TCGA-COAD was open-accessed, ensuring patient anonymity without risk of patient identification. All institutions contributing annotated biospecimens provided documentation to the TCGA, and have obtained ethical approvals to use the sample and data according to the human subjects protection and data access policies in TCGA program. Archival material derived from the AVANT-trial (BO17920) was performed in accordance with the declaration of Helsinki[30,31]. Protocol approval was obtained from the local medical ethics review committees or institutional review boards at participating sites.

### Study population

The TCGA-COAD dataset was used for training and extracting features and histologic patterns using SSL. This dataset consisted of 451 WSIs from 444 unique patients[29] with matched genetic and transcriptomic information. We excluded duplications and WSIs with erroneous resolution that were not suitable for the analyses (i.e. only several kilobytes in size). The final TCGA training set included 435 WSIs from 428 patients with a diagnosed pathological TNM-stage I-IV colon carcinoma (333 alive, 94 dead, and 1 missing vital status). We referred to the source population which TCGA-COAD representing as the "standard-of-care" group, contrasting it with the clinical trial external test data described below.

As external dataset we leveraged a study comprising 1213 colon cancer patients with available diagnostic H&E WSIs (one WSI per patient) as part of the clinical Bevacizumab-Avastin® adjuVANT

(AVANT) trial[30,31]. Bevacizumab, a VEGF monoclonal antibody, had initially been shown to improve the OS in patients with metastatic colon cancer when jointly used with the standard chemotherapy[33]. In the phase III AVANT trial, with an intent-to-treat population of 3451 patients, an open-label design was used[30]. Patients were randomly assigned in a 1:1:1 ratio to three different treatment regimens: FOLFOX-4 (intravenous 5-fluorouracil/folinic acid plus oxaliplatin), bevacizumab-FOLFOX4, and bevacizumab-XELOX (oral capecitabine plus intravenous oxaliplatin)[30]. The study aimed to investigate whether adding bevacizumab to the standard oxaliplatin-based adjuvant chemotherapy could improve DFS among patients with stage II-high risk and III colon cancer[30]. However, the trial was prematurely terminated due to the serious adverse effect on the patient's OS in the bevacizumab-treated group[30]. The AVANT trial was chosen also due to the previously studied potential correlation of VEGF, the stromal compartment (e.g. TSR), and patient prognosis[31,38,43]. For a detailed overview of the trial and patient characteristics, see Zunder et al.[31].

Given the unique treatment regimen of bevacizumab and its adverse effect in non-metastatic colon cancer, as also proven by a predecessing clinical trial, the NSABP protocol C-08 trial[63], we decided to primarily validate OS prediction, trained in the TCGA-COAD, within the control group who only received FOLFOX-4 (without bevacizumab) and refer to it as the "AVANT control group". Subsequently, as several phase III trials had demonstrated that FOLFOX and XELOX are comparable in the context of metastatic colorectal cancer[33–35], we thus combined the bevacizumab+FOLFOX-4 and bevacizumab+XELOX groups into a unified "AVANT-experimental group" and conducted a separate analysis to predict OS within the bevacizumab-treated patients.

## Data pre-processing

**Tissue segmentation and image tiling**. We used the preprocessing methods described in our previous study[64]. Tissue areas in WSIs were segmented against background at 10x magnification level (pixel size approximate 1.0 um). WSIs were divided into non-overlapping image tiles of size 224×224 pixels. The selection of a 10x magnification was based on two key considerations. Firstly, it aligns with the standard magnification utilized by pathologists during microscopic assessments in clinical practice. Secondly, we conducted visual inspections of tiles at 20x, 10x, and 5x magnifications, and found that tiles at 10x magnification provided an optimal balance of capturing sufficient detail while also offering a reasonably sized overview of the morphological structure. In addition, to overcome the variability of color stains from different scan facilities in the TCGA and AVANT cohorts, we further applied the color normalization[65]. In total, we obtained 1,117,796 tiles in the TCGA training set (i.e. TCGA-COAD), and 4,827,055 tiles in the AVANT external test set (AVANT-COAD, consisting of the standard and experimental treatment groups).

## Extracting image features using Barlow Twins

We trained the SSL Barlow Twins feature extractor based on 250,000 image tiles randomly selected from the TCGA-COAD dataset. The Barlow Twins extracted unique latent vectors (128 dimension) from the preprocessed image tiles in TCGA. The model is based on ResNet-like architecture consisted of several ResNet layers and one self-attention layer[23]. In essence, the Barlow Twins calculates the cross-correlation matrix between the embedding outputs of two identical twin networks, both fed with distorted versions of the same image tile[25]. It is optimized to make the correlation matrix close to the identity matrix[25]. We used the batch size of 64 trained on a single NVIDIA® Tesla V100 GPU for 60 epochs. The Barlow Twins feature extractor was frozen after the training and used to project image tiles into latent representation in the entire TCGA training set.

To facilitate the downstream analysis, we also applied five-fold CV partition in the TCGA-COAD training set on patient level balanced the

on AJCC TNM stage, OS outcomes (i.e. death or censor), and binned survival time categories.

## Leiden community detection algorithm

The Leiden clustering algorithm is a graph-based clustering algorithm that aims to identify distinct communities or clusters within graph data[66]. In brief, it optimizes the modularity function (Eq. 1), in such way to maximize the difference between the actual number of edges and the expected number of edges in a community[66]. This modularity function also includes a resolution parameter $\gamma$, with higher values leading to more clusters and lower values leading to fewer clusters. Leiden clustering is initiated by assigning each node in the graph to its own individual cluster, treating them as separate communities, then iteratively optimizes the modularity function by moving a node from its current cluster to a neighboring cluster or by merging clusters until the algorithm converged and revealing distinct communities or clusters. We employed the Leiden clustering in a particular workflow. We began by constructing a neighborhood graph using the K = 250 nearest neighbors from a pool of 200,000 randomly selected latent image vectors from a training set. Subsequently, we applied the Leiden algorithm to identify clusters within this neighborhood graph. These cluster labels were then propagated to each individual image tile across the entire dataset, once more utilizing the K-nearest neighbors approach.

$$\mathcal{H} = \frac{1}{2m} \sum_c \left( e_c - \gamma \frac{K_c^2}{2m} \right) \tag{1}$$

Modularity measures the difference between the actual number of edges in a community and the expected number of such edges. $e_c$ denotes the actual number of edges in community $c$ and the expected number of edges is expressed as $\frac{K_c^2}{2m}$, where $K_c$ is the sum of the edges of the nodes in community $c$ and $m$ is the total number of edges in the network. $\gamma$ is the resolution parameter, with higher values leading to more clusters and lower values leads to fewer clusters.

## Identification of HPCs

We identified HPCs as clusters obtained from the Leiden community detection algorithm operated on 128-dimensional image features extracted through the Barlow Twins encoder. The Leiden method was also used for quality control to eliminate artifacts and under-focused image tiles. Quality control was carried out after training the Barlow Twins and extracting image features. In the overall process, we initially generated a substantial number of Leiden clusters. Next, we visually examined sample image tiles from each cluster and removed clusters exhibiting artifacts such as air bubbles, foreign objects, etc., or those containing under-focused images. In particular, within the training set of a randomly selected CV fold (fold 0), we randomly selected 200,000 latent image vectors to generate Leiden clusters. We obtained 125 clusters at a high resolution of $\gamma = 6$. The Leiden labels were propagated into the entire TCGA set using again the K-nearest neighbors method. Next, we inspected randomly selected sample tiles ($N = 32$) from each cluster and identified 12 clusters containing predominately artifacts or underfocused images. Image tiles labeled by these 12 clusters were subsequently removed from the further analyses. The HPCs were newly derived in this cleaned dataset by re-running the Leiden clustering. Optimization of the HPCs was conducted using primarily unsupervised methods and secondarily confirmed using supervised methods. Importantly, both the two approaches converged on the same optimal Leiden resolution, as elaborated below.

## Optimization of HPCs using unsupervised methods

Leiden resolutions (i.e. $\gamma = 0.4, 0.7, 1.0, 1.5, 2.5,$ and $3.0$) were optimized using three unsupervised statistical tests: the Disruption score,

Silhouette score, and Daves-Boundin index. Due to the potentially high variance in data from the different institutions in the TCGA, all three scores were weighted by the mean percentage of the institution presence in each cluster. We consistently identified the optimal Leiden resolution as 1.5 through the three aforementioned statistical tests (Supplementary Fig. 1a).

## Optimization of HPCs and the prediction of OS using Cox regressions with L2 regularization

To identify optimal HPC configurations and their associations with patient OS, we trained L2 regularized Cox regressions for OS prediction using 5-fold CV. The Cox regressions were trained separately among standard-of-care colon cancer patients (i.e. TCGA-COAD) and among the AVANT-experimental group.

**OS prediction in the standard-of-care group.** Prediction of OS from HPCs among the standard-of-care group was trained within TCGA-COAD using 5-fold CV and tested in the independent AVANT control group. Specifically, for each CV fold within TCGA-COAD, we began by generating a range of Leiden clustering configurations at various resolutions, including γ values of 0.4, 0.7, 1.0, 1.5, 2.5, and 3.0, using the method described earlier. Next, at each Leiden resolution, we calculated the compositional representation of HPCs for each WSI (see Main Fig. 1c), followed by a center-log-ratio transformation (Eq. 2). This transformation was designed to mitigate interdependencies among HPCs, ensuring that the independence assumptions required for subsequent Cox regression analysis were met. The L2 regularized Cox regressions were then trained at the patient level, with one WSI per patient considered. At each Leiden resolution, we performed a multivariable L2 regularized Cox regression, incorporating all center-log-ratio-transformed HPCs specific to that resolution. We finetuned L2 regularizer (alpha) through an iterative process involving 50 steps, spanning the alpha range from $10^{-4}$ to $10^4$. This sequence of steps was repeated across all five cross-validation folds. The optimal Leiden resolution and L2 regularizer was selected based the CV C-index.

Through this optimization process, the optimal Leiden resolution was determined to be 1.5, and the L2 regularizer alpha was fixed at 0.1842. Of the note, this optimal Leiden resolution of 1.5 concurred with the result from the unsupervised approaches. An HPC-based classifier was determined by the median predicted hazard obtained in the TCGA-COAD. Once the Cox model is optimized, we evaluated the final model performance in the external test set consisting of the AVANT-standard care group. First, the 47 HPCs were integrated into the AVANT-standard care group by employing the K-nearest neighbors method (K = 250), where each AVANT tile's HPC label was determined based on the majority votes from its nearest neighbors in the TCGA training set. Next, the trained Cox model, with optimized regularization and parameter estimates for the HPCs, was then applied to the AVANT-standard care group to test the prediction of OS (Supplementary Fig. 3b).

Furthermore, employing the same CV method, we trained a clinical baseline model on OS in TCGA-COAD using L2 regularized Cox regression incorporating age, sex, TNM staging, and TSR as predictors. We observed a c-index of 0.58 (bootstrap 95% CI = 0.49-0.67) in the independent AVANT-standard care group and the model-based risk classifier did not reach the statistical significance level (Supplementary Fig. 3c). This baseline model illustrates a simulation of decision-making in clinical practice as control, using the most readily available and relevant clinical and demographic variables. Our HPC-based model outperformed this clinical baseline model. In addition, we explored whether the HPC-based classifier add additional prognostic value to the existing important clinical predictors. We fitted an ordinary multivariable Cox regression within the external AVANT-standard care group, including HPC-based classifier, sex, age, TSR, and AJCC TNM

stage (Main Fig. 4a).

$$\mathrm{clr}(x_i) = \ln\left(\frac{x_i}{g}\right), x_i = \frac{\text{counting of tiles in } c_i}{\text{total number of tiles in each WSI}}, g = \left(\prod_{i=1}^{n} x_i\right)^{\frac{1}{n}}$$

(2)

The center-log-ratio transformation (clr) calculates the natural logarithm of the ratio ($x_i$) of compositional data for a specific cluster ($C_i$) to the geometric mean ($g$) of the compositional data across all clusters. The compositional data ($x_i$) is obtained by dividing the counting of tiles in cluster $C_i$ by the total number of tiles in each WSI. The geometric mean ($g$) is computed by multiplying the compositional data values ($x_i$) for each cluster ($C_i$), ranging from $i = 1$ to $n$ (where $n$ is the total number of clusters), and raising the resulting product to the power of $\frac{1}{n}$.

**OS prediction in the AVANT-experimental group.** Bevacizumab, a unique intervention investigated in the AVANT trial, was unlikely accessed by patients from the TCGA-COAD cohort. Considering the significant poor prognostic outcome associated with bevacizumab in non-metastatic colorectal cancer, we postulated that the relationship between HPCs on OS might be influenced by this intervention. Furthermore, we hypothesized that estimates of HPCs on OS trained from the general COAD population, such as TCGA, might not be applicable to bevacizumab-treated patients.

To address these hypotheses, we conducted a separate 5-fold CV estimating the relationship between HPCs and OS within the bevacizumab-treated patients (AVANT-experimental group). Similarly, we generated 5-fold train-validation split stratified by TNM stage and survival time. We used the same sets of HPCs obtained at the previously optimized Leiden 1.5 resolution. Similarly, we modelled the center-log-transformed compositional data of HPCs on OS using Cox regressions with L2 regularization. We fine-tuned the L2 regularizer specifically for AVANT-experimental group. The model performance of HPCs on OS prediction was evaluated using c-index in the 5-fold CV validation sets (Supplementary Fig. 3d).

To understand the importance of each HPC on OS, we employed SHAP values[67]. SHAP values measure the marginal contribution of a HPC towards the predicted OS, considering all possible combinations of features. We highlighted top 20 important HPCs favoring and hindering survival for both standard-of-care and AVANT-experimental treated COAD patients.

## Interpretation of HPCs
**Plotting HPCs using UMAP and PAGA plot.** We applied UMAP dimensionality reduction[68] to TCGA-COAD tile vector representations (128 dimensional vectors), color-coded by 47 HPC IDs using optimized Leiden clustering configuration from the prior step. Next, we generated a PAGA plot where each HPC is a node connected by lines based on vector similarity. Pie charts within each HPC node showed annotated tissue type percentages annotated independently by three experts (ASLPC, JHJMvK, and MP) (see below). We analyzed this plot to identify and describe significant interconnected clusters (Main Fig. 2b). Based on the PAGA plot, we also defined the "super-clusters" (Main Fig. 2c) according to the interconnections among HPCs and tissue composition.

## Pathologist assessment of HPCs
**Histopathological assessment and characterization of HPCs.** The histomorphological features of 47 HPCs derived in TCGA training set were comprehensively and independently analyzed by two expert pathologists (ASLPC and JHJMvK) and one medical researcher (MP). The analysis was exclusively conducted using image tiles and the assessors were kept blinded to results from other analyses.

Specifically, we randomly selected a set of 32 tiles from each HPC within TCGA and each individual tile was examined with specific focus on tumor epithelium, tumor stroma, and immune cells. Attention was also paid to general tumor differentiation grade, tumor stromal amount, stromal classification, i.e. aligned or disorganized, stromal neovascularization or other notable patterns (e.g. dysplastic tissue, fatty tissue, muscle tissue fibers, blood vessels, erythrocytes, etc.)[3,43,59]. Each assessor evaluated the tissue composition based on 32 randomly selected tiles per HPC, providing an average tissue composition for each HPC. In cases of conflicting tissue annotations, the assessors carried out discussions to achieve consensus. In addition, a short label was given to each HPC based on tissue annotation either according to the first and second predominant tissue types and patterns (e.g. "aligned stroma - well differentiated tumor epithelium"), or as the single most dominant tissue type covering ≥70% area of an average tile (e.g. "necrotic"). The tissue description, composition, and short labels of all 47 HPCs were displayed in Supplementary Table 1.

Assessment of HPC consistency within and across TCGA and AVANT cohorts We performed various qualitative and quantitative analyses to evaluate the within-cluster and between-cluster heterogeneity of HPCs, as well as their transferability from TCGA to AVANT cohorts. Initially, qualitative visual assessments were independently conducted by three experts (ASLPC, JHJMvK, and MP) to evaluate the concordance within and discordance between HPCs. This evaluation followed the established protocol for tissue type analysis, utilizing the previously randomly selected 32 tiles per HPC. The assessors noted consistent histomorphological patterns within HPCs and diverse patterns across various HPCs.

In addition, we conducted quantitative objective tests within TCGA and AVANT tiles separately to determine if the learned morphological features of the HPCs could be replicated by human eyes. We displayed three rows, each containing five tiles, which is referred to as a "question". Among these rows, two belonged to the same HPC, while the third row, also referred to as the "odd HPC", were tiles randomly selected from a different HPC. All example tiles were randomly selected within each HPC. The researcher (MP) was required to identify the "odd HPC", and upon doing so, the next question was presented (example shown in Supplementary Fig. 2). In order to calculate the success rate, we ran 50 questions per HPC and we repeated the experiment for all 47 HPCs. The test was conducted on a closed website accessible through login and was conducted separately for tiles randomly selected from TCGA and AVANT. The conducted experiment exhibited an average completion time of under 10 seconds per question. The success rate was determined for each HPC by dividing the number of incorrect answers by the total number of questions (50 questions per HPC). We hypothesized that most HPCs would be distinguishable. However, some HPCs may be challenging to differentiate due to tissue similarities, either within the same super-cluster or through similar tissue morphology (e.g., muscle tissue and tumor stroma). The results showed that HPCs in close proximity to each other in the PAGA or belonging to the same super-cluster were indeed more prone to being mistaken.

To validate whether HPCs derived from the TCGA training set generalized well in the external AVANT test set, three assessors (ASLPC, JHJMvK, and MP) independently carried out visual examination of tissue characteristics in 32 randomly selected tiles from each HPC in both the AVANT trial and the TCGA set, comparing their histomorphological features regarding tissue types and composition established in the previous step. Assessors confirmed that the consistent and meaningful patterns, as learned by the Barlow Twins, could be replicated across other clinical cohorts like the AVANT trial. To validate our observation, we compared the objective test results between TCGA and AVANT, focusing on the overlap in both misclassified and correctly classified HPCs across the datasets.

## Linkage between survival-related HPCs, immune landscape, and gene expression data

We calculated Spearman correlations between HPCs and RNASeq-derived immune features from TCGA-COAD data, correcting for false discovery rate (0.05) using Benjamini-Hochberg correction. We created bi-hierarchical clusters for immune features and top 20 significant HPCs linked to OS in both standard-of-care and AVANT-experimental groups. Clustering was based on correlation coefficients using Euclidean distance and Ward's aggregation.

In addition, we performed GSEA to explore the potential associations between HPCs and the major cancer hallmark pathways, as previously described[69]. The analysis was conducted within 282 TCGA-COAD patients with available gene expression data for 20,530 genes. For each HPC, we calculated the Spearman correlation coefficients with the expression data of these 20,530 genes and sorted the resulting correlation coefficients to rank the genes in a descending order. Subsequently, enrichment scores for 50 major cancer hallmark pathways, as defined by the "MSigDB_Hallmark_2020" gene set[58], were computed for each HPC based on the generated ranked gene list. A positive enrichment score indicated higher composition of a HPC was associated with gene enrichment in a pathway and a negative value suggested higher composition of a HPC was associated with under-representation of a pathway. We plotted the cancer hallmark enrichment scores for the top 20 important HPCs related to OS, for both the general and experimental treatment groups. We set a significance level of 0.01 of the false discovery $q$ value. The analysis was conducted using the gseapy (1.0.4) Python package. Two-way hierarchical clustering was based on the Euclidean distance of the enrichment scores and Ward's aggregation method.

Lastly, we created a Supplementary Table 2 to grant a detailed overview of the significant HPCs associated with OS, including the direction of the association (worse or better OS) for both treatment groups (TCGA and AVANT standard, or AVANT-experimental) and in-depth histopathological analyses potentially explaining these associations with appropriate references.

### Reporting summary
Further information on research design is available in the Nature Portfolio Reporting Summary linked to this article.

## Data availability
The publicly available TCGA-COAD can be accessed at Genomic Data Commons portal (https://gdc.cancer.gov/). The immune landscape signatures for TCGA-COAD is available in Thorsson et al.[70]. The AVANT data that support the findings of this study are available from Genentech Inc., Roche. However, access to these data is restricted as they were used under license for the current study and are not for commercial use to protect patient privacy. Data may however be available from the authors upon request for research purposes with permission from Genentech Inc., Roche. Data analyses are based upon publicly available Python software packages and codes are available from our previous publication[23] (https://github.com/AdalbertoCq/Histomorphological-Phenotype-Learning). The remaining data are available within the Article, Supplementary Information or Source Data file. Source data are provided with this paper.

## Code availability
Source code and Python software packages are available from our previous publication[23] (https://github.com/AdalbertoCq/Histomorphological-Phenotype-Learning).

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

## Acknowledgements

The work is supported by the Swedish Research Council (BL, 2019-06360), NCI/NIH Cancer Center Support Grant (AT, P30CA016087), the Stichting Fonds Oncologie Holland and the Bollenstreekfonds, Hillegom, Netherlands (WEM, no grant numbers). We thank the team of NYU Langone High Performance Computing (HPC) Core's resources supporting us to perform the analysis. We would like to thank the NYU Applied Bioinformatics Laboratories (ABL) for providing bioinformatics support and helping with analysis of the data. ABL is a shared resource partially supported by the Cancer Center Support Grant P30CA016087 at the Laura and Isaac Perlmutter Cancer Center. We would also like to thank Leslie Solorzano for her input in data preprocessing. The AVANT-trial (BO17920) was originally financed by Genentech Inc., Roche, Switzerland. The authors thank Nikolas Jan Rakebrandt as liaison from Roche.

## Author contributions

N.C. and A.C.Q. contributed equally as the co-second authors. B.L. executed data processing, model training, and data analyses. B.L., N.C., A.C.Q. wrote python codes used to run all the experiments. M.P. and W.E.M. provided coordination in obtaining H&E slides and corresponding data from Bevacizumab-Avastin® adjuVANT trial. M.P., A.S.L.P.C., J.H.J.M.v.K., A.K. provided histological assessment and interpretation of HPCs. B.L. and M.P. wrote the first draft of the manuscript which was later revised and approved by all co-authors. B.L., N.C., A.C.Q., T.S., H.L., A.T. provided expertise in deep learning, biostatistics, and bioinformatics. M.P., A.S.L.P.C, J.H.J.M.v.K, R.A.E.M.T, and W.E.M provided expertise in colon cancer histopathology and biology. A.T. supervised the study.

## Competing interests

AT is a co-founder of Imagenomix. The remaining authors declare no competing interests.
