## [Transparent Peer Review file · Nature Communications]

Self-Supervised Learning Reveals Clinically Relevant Histomorphological Patterns for Therapeutic Strategies in Colon Cancer

Corresponding Author: Professor Aristotelis Tsirigos

Version 0:

Reviewer comments:

Reviewer #1

(Remarks to the Author)

The paper is well written and presented, the approach is interestingly novel, and the application is time critical and cutting edge. Though I am in favour of this paper to be considered for publication, I have two main concerns that need to be addressed first, to establish the state-of-the-art.

1) Google has recently made available their self-supervised large foundation model for digital pathology, that is now available for use to the public through an API, hence why not use that for the feature embedding and then just use an interpretable mechanistic lightweight model for the downstream task at hand?

blog: <https://blog.research.google/2024/03/health-specific-embedding-tools-for.html>

paper: <https://arxiv.org/abs/2310.13259>

code API: https://docs.google.com/forms/d/1auyo2VklzuiAXavZy1AWUyQHAqO7T3BLK-7ofKUvug/viewform?edit_requested=true#question=1168037695&field=173852432

2) As a computer scientist, I am not completely convinced regarding the choice of Barlow Twins model for this application, especially since Barlow Twins was proposed to be specifically INVARIANT to certain feature artefacts... through this is helpful in structured applications to avoid redundant features, but cancer is so heterogeneous! How do you know that some specific feature to a certain tumour sub-type would not be treated as outlier, and hence detract from personalised predictions?

3) Again, connected to the last point, a statistical evaluation of the feature invariance and its clinical alignment is needed... for example is it possible to tweak the Barlow Twins to leak some features and then note whether those were actually redundant or not?

(Remarks on code availability)

I have reviewed the code in the sense that it provides solid documentation as well as visualisation of the results on github itself even without installing and running the code itself locally, which I did not have time to do.

Reviewer #2

(Remarks to the Author)

The authors present an application of their previously published work, histomorphological phenotype clustering (HPC) to Colon Adenocarcinoma datasets.

The authors show that by leveraging self-supervised learning, it is possible to learn phenotypically distinct clusters of image tiles and then use these clusters in downstream analysis to address specific biological questions, such as links to overall survival (OS).

The manuscript is well written and the overall approach has been shown to work. I have a few questions I would like to see

addressed:

1. Scholarship: There are more self-supervised approaches and applications that more closely resemble the authors work which have been omitted. Specifically Cisternino et al., 2023 - <https://www.biorxiv.org/content/10.1101/2023.08.22.554251v1.full>

2. The HPCs are mapped back onto the WSI and demonstrate significant structure and clustering, as one would expect if they represent key histological concepts. However, the authors do not assess quantitatively and robustly how well a given HPC recovers the entire structure from a WSI. I think this is important to address, because otherwise, there's no way of seeing how much is missed or mis-classified. To do this, I would suggest getting full ground truth annotations of tissue substructures and assessing the accuracy/loY of tiles belonging to a given HPC, recovering that substructure. This would give weight to HPC proportions being truly meaningful.

3. Many HPCs correlate / align with known risk factors for cancer, such as immune cell fraction. Are there any HPCs that have effects of OS that are truly novel and don't correlated with currently known risk factors. I understand the authors have demonstrated overall that HPCs add predictive power over baseline measures, but the interpretation is key.

4. Figure 5:

- Infiltrated stroma / moderately differentiated tumour are associated with better survival
- Longitudinal muscle fibres or vessel-like stroma are associated with poor survival?

How do I interpret this, because those are counterintuitive to me. Surely normal tissue histology and the plane it was cut in have no bearing on survival and certainly not a negative one.

(Remarks on code availability)

The code is poorly structured, with lots of .py scripts sitting in the main repo and not organised. They provide a comprehensive README.md.

Reviewer #3

(Remarks to the Author)

The present manuscript entitled "Self-Supervised Learning Reveals Clinically Relevant Histomorphological Patterns for Therapeutic Strategies in Colon Cancer" by Liu et al. is a well-written and concisely presented study.

1. Strengths of the study

- a) Interdisciplinary team
- b) Test set and validation set
- c) Excellent visualisation of the study results
- d) Figure 7 highlights the potential clinical implication very well

2. Limitations of the study

a. My major points are already addressed by the authors in the discussion

b. In my opinion an additional paragraph should be added at the end of the discussion trying to address the following points:

- How should Institutes of Pathology concretely implement this approach?
- First report traditionally, then apply the proposed method in terms of additional information?
- Or shall this approach replace the standard reporting?
- Do the authors plan a validation study including more centers?
- How can the obtained results be correlated with the actual parameters proposed by the ICCR?

(Remarks on code availability)

Version 1:

Reviewer comments:

Reviewer #1

(Remarks to the Author)

The authors have fully addressed my comments and I am now satisfied for the revised manuscript to move towards publication.

(Remarks on code availability)

Reviewer #2

(Remarks to the Author)

I thank the authors for addressing all of my concerns.

(Remarks on code availability)

Reviewer #3

(Remarks to the Author)

The points raised in the review have been addressed satisfactorily by the authors.

(Remarks on code availability)

Response to Reviewers

Reviewer #1 - ML, image analysis:

Comment 1: The paper is well written and presented, the approach is interestingly novel, and the application is time critical and cutting edge. Though I am in favor of this paper to be considered for publication, I have two main concerns that need to be addressed first, to establish the state-of-the-art.

Response: *First, we would like to thank the reviewer for choosing to review our paper. We appreciate their compliments on the form and contents of our paper, as well their acknowledgement on the importance of the subject. Next, we will address each of the understandable concerns raised by the reviewer.*

Comment 2: 1) Google has recently made available their self-supervised large foundation model for digital pathology, that is now available for use to the public through an API, hence why not use that for the feature embedding and then just use an interpretable mechanistic lightweight model for the downstream task at hand?

blog: <https://blog.research.google/2024/03/health-specific-embedding-tools-for.html>

paper: <https://arxiv.org/abs/2310.13259>

code API:

https://docs.google.com/forms/d/1auyo2VklzuiAXavZy1AWUyQHAqO7T3BLK-7ofKUvug/viewform?edit_requested=true#question=1168037695&field=173852432

Response: *The reviewer makes a good point and we agree that in the future, foundation models can potentially be used as a first step in similar studies. Recently, foundation models, including our own HP-Atlas¹, have been shown to perform well in high-level downstream tasks, e.g. cancer subtyping. However, whether these pan-cancer models can identify consistent and interpretable histomorphological patterns at high resolution in any given tissue type is still unclear. This is why we chose to train a self-supervised (SSL) model specifically on colon cancer images. It would be very interesting to compare tissue-specific SSL models to pan-cancer models, but we believe it is beyond the scope of this work to do this systematically. Nevertheless, since this is a very interesting question, we decided to proceed with some initial analyses to quantify the concordance between these approaches.*

Specifically, we applied two pre-trained foundation models, UNI² and HP-atlas¹, to convert tile images to embeddings in the training set of TCGA colon adenocarcinoma (TCGA-COAD) and in the AVANT (Bevacizumab-Avastin®adjuVANT trial) external test set. Next, following a similar analysis pipeline as outlined in the manuscript, we performed Leiden clustering based on the UNI and HP-atlas features in TCGA-COAD. We selected the Leiden configurations for the two

foundation models based on the number of clusters obtained, aiming to match as closely as possible the clusters generated by our Barlow Twins-COAD model.

Then, we calculated the Adjusted Mutual Information (AMI)³ to evaluate the consistencies between COAD-BarlowTwins clusters and clusters derived from the two foundation models. The AMI between Barlow Twins-COAD and HP-Atlas-based clusters was estimated as 0.432, the AMI between Barlow Twins-COAD and UNI-based clusters was estimated as 0.387.

Additionally, we evaluated the consistency of the histologic patterns identified by the two foundation models by conducting the pathologist evaluation. In brief, the assessor was shown three groups of image tiles, each containing five tiles. Two groups were from the same HPC, and the third was from a randomly selected other HPC, also called the "odd HPC". The assessor was required to identify the "odd HPC". Each HPC underwent 50 tests to determine the success rate (see Online Methods: Pathologist assessment of HPCs, for details). The pairwise comparisons of test accuracies from the three models are shown in Fig. 1d-f. Wilcoxon signed-rank tests revealed no significant difference in test accuracy between Barlow-Twins-COAD-based and HP-atlas-based clusters ($p=0.159$). However, Barlow-Twins-COAD-based clusters demonstrated significantly better test accuracy compared to UNI-based clusters ($p=0.0217$).

Figure 1. Comparison of objective pathologist tests in BarlowTwins-based, UNI-based and HP-Atlas-based clusters. (a-c) Test results for BarlowTwins-based, UNI-based clusters

and HP-Atlas-based clusters in the AVANT study. Green bars indicate the percentage of successfully identified odd clusters. (d-f) Box plots and Wilcoxon signed-rank tests show pairwise comparisons in success rates between Barlow-Twins-COAD and HP-Atlas(d), Barlow-Twins-COAD and UNI (e), as well as HP-Atlas and UNI models (f).

Lastly, we assessed how well the clusters derived from the two foundation models predicted overall survival in the AVANT external test set. The prediction performances from the UNI-based clusters (c -index=0.60, log-rank p -value=0.022) and HP-Atlas-based (c -index=0.64, p -value=0.055) in the AVANT external test results were shown in Fig. 2a and b. Our conclusion is that HP-Atlas and UNI foundation models do not outperform the overall survival (OS) prediction performance achieved from the Barlow Twins-COAD HPC (c -index = 0.65, p -value = 0.005).

Figure 2. OS prediction performance in the external AVANT test set. (a) UNI-based OS prediction; (b) HP-Atlas-based OS prediction.

Comment 3: As a computer scientist, I am not completely convinced regarding the choice of Barlow Twins model for this application, especially since Barlow Twins was proposed to be specifically INVARIANT to certain feature artifacts... though this is helpful in structured applications to avoid redundant features, but cancer is so heterogeneous! How do you know that some specific feature to a certain tumour sub-type would not be treated as outlier, and hence detract from personalized predictions?

Response: We appreciate the reviewer's comment about invariance to certain feature artifacts, we believe this is an interesting and important point to clarify.

Before doing so, in order to avoid any confusion, we want to first clarify that the invariance property of Barlow Twins is not used to "avoid redundant features". In fact, the loss function has two terms, one for invariance and a different one for redundancy. It seems that this comment is about invariance and the next comment is about redundancy, so we will first address the invariance comment here.

As the reviewer points out, an invariance term is used in the loss function to favor embeddings that are invariant to certain artifacts. The reviewer raises an interesting point, wondering whether enforcing invariance may eliminate features that are histologically/biologically meaningful in heterogeneous tumors. To address this concern, we first point out that our model enforces invariance to properties that are not relevant to cancer heterogeneity. Specifically, we *applied standard crop, rotation, flip, and color distortion as suggested by Barlow Twins⁵, which are considered to be irrelevant to core features of histopathology images. Indeed, similar data augmentations, including color distortion, have been widely used in other networks, such as DINO-v2 ⁴ applied in the UNI foundation model. Furthermore, to address any remaining doubts, we reference our recent work where we used Barlow Twins to map the landscape of histomorphological phenotypes in lung adenocarcinoma¹. In this study, we showed that using the embeddings from Barlow Twins, we can derive clusters of histomorphological phenotypes that recapitulate all known simple and complex histologic patterns in lung adenocarcinoma (LUAD), suggesting that the full spectrum of heterogeneity is captured even within a specific cancer sub-type (i.e. LUAD).*

Comment 4: Again, connected to the last point, a statistical evaluation of the feature invariance and its clinical alignment is needed... for example is it possible to tweak the Barlow Twins to leak some features and then note whether those were actually redundant or not?

Response: *It seems that this comment is related to redundancy and not invariance. The redundancy term in the loss function encourages the learned features to be decorrelated, meaning that different features should capture different aspects of the data. The redundancy term is weighted by the "lambda" coefficient, with a smaller lambda leading to models learning features with higher redundancy. Initially, we set the "lambda" value to 0.05 when training the model as recommended in the Barlow Twins paper⁵. Following the reviewer's suggestion, we experimented with "extreme" values of the "lambda" weights to explore the impact of allowing more redundancy in the learned features (Fig 3). To this end, we retrained our models from scratch with new lambda values and repeated the entire computational analyses. We hypothesized that increased feature redundancy would result in fewer histomorphological phenotype clusters (HPCs) and potentially capture fewer meaningful histopathological structures. In line with our hypothesis, we indeed observed fewer clusters from the lambda-0.001 (N=42) and lambda-0.00001 (N=41) models as compared to the original lambda-0.05 model (N=47) (Fig. 3). This analysis suggests that when lambda is decreased, and as a result the number of redundant (i.e. highly correlated) features is increased, there is an impact, as expected, on the number of identified histologic patterns (HPCs). This observation highlights the usefulness of the redundancy term.*

Figure 3. Experiments with various lambda values for feature redundancy. (a) Relationship between lambda hyperparameter values and number of resulting HPCs. (b) OS prediction from lambda=0.001 model in the external AVANT cohort. (c) OS prediction from lambda=0.00001 model in the external AVANT cohort

Reviewer #1 (Remarks on code availability):

Comment 5: I have reviewed the code in the sense that it provides solid documentation as well as visualisation of the results on github itself even without installing and running the code itself locally, which I did not have time to do.

Response: We thank the reviewer for the positive comments on code documentation and the visualization of the results.

Reviewer #2 - ML, image analysis:

Comment 1: The authors present an application of their previously published work, histomorphological phenotype clustering (HPC) to Colon Adenocarcinoma datasets.

The authors show that by leveraging self-supervised learning, it is possible to learning phenotypically distinct clusters of image tiles and then use these clusters in downstream analysis to address specific biological questions, such as links to overall survival (OS).

The manuscript is well written and the overall approach has been shown to work. I have a few questions I would like to see addressed:

Response: *We would like to first thank the reviewer who has put time and effort in reviewing our study. We appreciate the recognition of our work and intentions, and are thankful for the compliment on how our manuscript is written. Next, we address each of the reviewer's questions below.*

Comment 2: Scholarship: There are more self-supervised approaches and applications that more closely resemble the authors work which have been omitted. Specifically Cisternino et al., 2023 - <https://www.biorxiv.org/content/10.1101/2023.08.22.554251v1.full>

Response: *As it is indeed a quickly developing field, we are grateful for this suggested reference and added it in our introduction. Moreover, we have extended our literature search to include more recent work published on self-supervised approaches and applications in our manuscript.*

Comment 3: The HPCs are mapped back onto the WSI and demonstrate significant structure and clustering, as one would expect if they represent key histological concepts. However, the authors do not assess quantitatively and robustly how well a given HPC recovers the entire structure from a WSI. I think this is important to address, because otherwise, there's no way of seeing how much is missed or mis-classified. To do this, I would suggest getting full ground truth annotations of tissue substructures and assessing the accuracy/IoY of tiles belonging to a given HPC, recovering that substructure. This would give weight to HPC proportions being truly meaningful.

Response: *The reviewer proposes a comparison to ground-truth annotations. To this end, we randomly selected 50 whole-slide images from the external dataset (AVANT) and annotated them by researcher MP manually in detail using QuPath software version 0.1.2. Annotations were drawn using standard pathologist labels based on the following tissue types: healthy and dysplastic tissue, necrosis, mucinous, muscle tissue, immune cells, fatty tissue, tumor stroma, and tumor epithelium. Annotations were checked by pathologists afterwards. We then assessed the performance of HPCs in terms of identifying correctly these manual annotations as recommended by the reviewer. First, annotation masks were generated for each manual annotation label and overlapped at the tile level on each of the 50 slides. An annotation label was assigned to a tile if at least 50% of the tile area was covered by the corresponding annotation. We then compared the HPC super-cluster tile labels with ground-truth tile labels ($N = 7,874$ tiles) and calculated a macro-AUC of 0.830 (95% bootstrap confidence intervals [CI] = 0.826-0.834, with 1000 iterations) and a micro-AUC of 0.931 (95% bootstrap CI: 0.927-0.935).*

To illustrate the excellent concordance between SSL-assigned labels and manual annotations, we plotted three randomly selected example heatmaps of pathologist tissue annotations and corresponding predicted labels formed by HPCs (Figure 4). The tissue labels predicted by HPCs (right) consistently reflect the pathologist annotated tissue areas in H&E-stained tissue slide (left).

Figure 4. HPC-based tissue areas (right) versus pathologist annotated tissue areas on H&E-stained tissue slide (left) in three separate colon carcinoma cases (A-C).

Comment 4: Many HPCs correlate / align with known risk factors for cancer, such as immune cell fraction. Are there any HPCs that have effects of OS that are truly novel and don't correlate with currently known risk factors. I understand the authors have demonstrated overall that HPCs add predictive power over baseline measures, but the interpretation is key.

Response: *We greatly appreciate the reviewer's recognition of our work on correlating HPCs with cancer risk factors and acknowledging that our HPCs provide additional predictive potential beyond a clinical baseline model (trained on age, sex, tumor-stroma ratio (TSR), and AJCC stage; see Supplementary Figure 3C).*

To directly address the reviewer's point regarding interpretations, we have again delved into the literature and had more extensive discussions with expert pathologists in trying to ascertain potential explanations of this effect on patient-related outcomes per HPC, as we highlight in 'Results' but predominantly expand on in-depth in the 'Discussion'.

We have therefore created a whole new table, Supplementary Table 2, linking HPCs to major immune and oncogenic pathways and interpretations how these associations could influence patient survival as well as their response to the experimental treatment with Bevacizumab and standard chemotherapy. As the reviewer points out, this was done by referring to current literature and knowledge. We identified certain morphologies that are not typically analyzed in pathology, such as the amount or organization of tumor stroma (e.g., HPC 0, 40), neovascularization (HPC 11, 21), and tissue composition likely corresponding to microsatellite instability status (HPC 26, 46). Additionally, we found novel characteristics that have not yet been well-described, such as necrosis and VEGFa gene enrichment (HPC 5, 18) in relation to response to Bevacizumab. To validate whether these novel markers are truly linked to novel tumor biology, future studies involving experimental studies on molecular, cellular, and animal models as well as validations on population-based studies are warranted.

Comment 5: Figure 5:

- Infiltrated stroma / moderately differentiated tumour are associated with better survival
- Longitudinal muscle fibres or vessel-like stroma are associated with poor survival?

How do I interpret this, because those are counterintuitive to me. Surely normal tissue histology and the plane it was cut in have no bearing on survival and certainly not a negative one.

Response: *First, we would like to thank the reviewer for their effort to interpret the sometimes seemingly counterintuitive findings. In response to this comment and to the previous question, we have created the new Supplementary Table 2, which includes all HPCs that are associated with OS either in the experimental treatment or standard treatment groups (as shown in manuscript Figure 5). In addition to our Supplementary Table 1 where we described tissue*

composition of each HPCs, Supplementary Table 2 specifically highlights and summarizes features and patterns in those HPCs that have been previously linked to OS in the literature (references included in table). Additionally, we have revised the Discussion section to include these findings and added extra references. Below, we have added the specifically mentioned associations the reviewer requested clarification of.

As the reviewer points out, our model indicates that the HPC labels of "Infiltrated stroma / moderately differentiated tumour are associated with better survival". First, to clarify, the labels of the HPCs are short titles, summarizing the main feature shown in that HPC, facilitating use of all HPCs in general (e.g. in figures), however, there are often more features which potentially associate with OS than only those used in the label description. With 'infiltrated' stroma, we meant 'infiltrated by immune cells, or a high influx of immune cells', for instance like HPC 13. This is a sign of a good host response against the tumor, which is a favorable outcome and there are many biomarkers known that correspond with this (e.g. Immunoscore). The well-moderately differentiated tumor epithelium (also known as 'low grade' tumor) is, in comparison to poorly-undifferentiated tumor (or high grade), also a known prognostic favorable factor, even more so with a combination of favorable factors, like illustrated by HPC 4. This is supported by the explanations and references we present in our novel Supplementary Table 2.

The reviewer also points out that our model indicates that "Longitudinal muscle fibres or vessel-like stroma are associated with poor survival". A possible explanation for the longitudinal muscle fibers, is that these also are very similar to stromal strands, indicating higher proportions of tumor stroma. Another possibility is for HPC 33 for instance, that we saw tumor buds invading the muscle tissue and/or serosa of the colon, a sign of a more aggressive tumor. Since the majority of the tiles consist of muscle fibers, the HPC is merely labeled 'Muscle tissue (longitudinal fibers)', although the association on OS is poor, which can indeed seem counterintuitive. Furthermore, vessels in stroma often indicate neovascularization, an active tumor remodeling its microenvironment and signaling for more nutrients to maximize potential invasion. The 'Vessel-like stroma/muscle' HPC 24, on the other hand, is a bit more difficult and shows different OS associations per treatment group. After extensive literature searching and with expert opinions, we summarize the evidence in Supplementary Table 2.

Regarding the non-cancerous tissue and a potential influence on outcome, there is an effect known as 'cancer field characterization' or 'field effect', in which the tissue surrounding the tumor also is affected by the storm of interleukins and cytokines, immune cells and active remodeling of the microenvironment⁶. Another potential explanation is that, as we model the fractional data of HPCs per slide per patient, higher fraction of non-cancerous tissue may indicate lower fraction of cancerous tissue in a certain slide, suggesting a potentially smaller tumor. Small tumor size, as also for instance stated in the TNM classification (lower T stage), is a known prognostic factor favoring survival.

Finally, we would like to refer the reviewer to our novel, detailed overview of the associations to OS per HPC in Supplementary Table 2, where these explanations are summarized, including references.

Reviewer #2 (Remarks on code availability):

Comment 6: The code is poorly structured, with lots of .py scripts sitting in the main repo and not organised. They provide a comprehensive README.md.

Response: *We appreciate this feedback and are grateful that the reviewer took the time to look through our codebase. We have provided a comprehensive and structured README file (<https://github.com/AdalbertoCq/Histomorphological-Phenotype-Learning/blob/master/README.md>) detailing each step required to execute the models and scripts.*

Reviewer #3 - Colon cancer, histopathology:

Comments 1: The present manuscript entitled "Self-Supervised Learning Reveals Clinically Relevant Histomorphological Patterns for Therapeutic Strategies in Colon Cancer" by Liu et al. is a well-written and concisely presented study.

1. Strengths of the study
 - a) Interdisciplinary team
 - b) Test set and validation set
 - c) Excellent visualisation of the study results
 - d) Figure 7 highlights the potential clinical implication very well

Response: *We are thankful for all the compliments and the recognition of the strengths of our work by the reviewer.*

Comments 2: 2. Limitations of the study

- a. My major points are already addressed by the authors in the discussion
- b. In my opinion an additional paragraph should be added at the end of the discussion trying to address the following points:
 - How should Institutes of Pathology concretely implement this approach?
 - First report traditionally, then apply the proposed method in terms of additional information?
 - Or shall this approach replace the standard reporting?
 - Do the authors plan a validation study including more centers?
 - How can the obtained results be correlated with the actual parameters proposed by the ICCR (International Collaboration on Cancer Reporting)?

Response: We thank the reviewer for acknowledging that our Discussion already addressed the major points. To address the reviewer's comments about potential clinical implementation and future validations, we have now indeed revised the Discussion and included an additional paragraph as suggested by the reviewer, as here below:

"Building upon the aforementioned findings, this study showcases the prospective clinical utility of AI-generated HPCs (Figure 7). Cancer WSIs were preprocessed into image patches and subsequently used to train SSL encoders and to form HPCs. These HPCs serve as condensed representations of the original WSIs, ready to be inspected by pathologists and enabling flexible linkage to multimodal omics data. These HPCs hold promise in classifying various tumor characteristics, potentially predicting patient prognosis and discerning distinct sensitivity groups to various therapies. Although this study was already trained on multicenter TCGA data and validated in an external, clinical multicenter cohort, we plan to conduct additional validations in other population-based external cohorts to strengthen its clinical applicability. In a pathology lab, this could be implemented as follows: Alongside the routine pathology report, an AI-generated report would provide personalized prognostic risk quantification (e.g., based on the patient's HPC composition and SHAP values). The report would also include tissue composition descriptions for each HPC, granting pathologists a complete overview per patient. Subsequently, with these reports used in the multidisciplinary team meetings, a colon cancer patient can be granted an optimal personalized treatment strategy."

Regarding the actual correlation to the mentioned ICCR parameters, we would first like to refer the reviewer to our Supplementary Table 1, where we have described all HPCs in detail. We will highlight some here, of which we have also added some to our Discussion:

Given the significant potential implications of these HPCs on OS and therapy response, implementation of this AI-based analysis may be advocated to international guideline organizations, such as the TNM evaluation committee of the Union for International Cancer Control (UICC). Our HPC-based analysis not only summarizes the TNM classification, but also correlates to parameters assessed in standard pathology diagnostics, such as the International Collaboration on Cancer Reporting (ICCR)⁷. Of note, ICCR parameters pertain to a global assessment on whole slide or patient level, while HPCs can provide a local assessment of histopathological characteristics on the tile level. However, these tile level histopathological characteristics can still be linked to global ICCR parameters. For example, a core element of the ICCR, 'Histological tumor grade', can be correlated to HPCs containing tumor epithelium with different differentiation statuses. HPCs 2 and 3 are for instance characterized by well-moderately differentiated (or 'low-grade') tumor epithelium, while HPC 45 is formed by poorly-to-undifferentiated (or 'high-grade') tumor epithelium-containing tiles. Moreover, 'Histological tumor type' in the ICCR can be linked to HPC 12, which contains predominantly mucinous tumors. Lastly, the TNM classification, and particularly tumor size, in the ICCR reporting guidelines may be linked to HPCs containing healthy colon tissue (e.g. HPC 39), since WSIs with more of the healthy HPC 39, may correspond to a fractionally smaller tumor size (i.e. lower T-stage)."

Response to editor and reviewers - Liu et al. 'Self-Supervised Learning Reveals Clinically Relevant Histomorphological Patterns for Therapeutic Strategies in Colon Cancer' (**NCOMMS-24-11291A**)

References

1. Claudio Quiros et al. Mapping the landscape of histomorphological cancer phenotypes using self-supervised learning on unannotated pathology slides. Nat. Commun. 2024
2. Chen et al. Towards a general-purpose foundation model for computational pathology. Nat. Med. 2024
3. Romano et al. Adjusting for Chance Clustering Comparison Measures. J. Mach. Learn. Res. 2016
4. UNI/uni at main · mahmoodlab/UNI. GitHub
5. Zbontar et al. Barlow Twins: Self-Supervised Learning via Redundancy Reduction. BioRxiv 2021.
6. Lockhead et al. Etiologic field effect: reappraisal of the field effect concept in cancer predisposition and progression. Mod. Path. 2015
7. Loughrey et al. Colorectal Cancer Histopathology Reporting Guide. 1st edition. International Collaboration on Cancer Reporting; Sydney, Australia. 2020. ISBN: 978-1-922324-01-6.

Response to Reviewers

Reviewer #1 - ML, image analysis:

Comment 1: The paper is well written and presented, the approach is interestingly novel, and the application is time critical and cutting edge. Though I am in favor of this paper to be considered for publication, I have two main concerns that need to be addressed first, to establish the state-of-the-art.

Response: *First, we would like to thank the reviewer for choosing to review our paper. We appreciate their compliments on the form and contents of our paper, as well their acknowledgement on the importance of the subject. Next, we will address each of the understandable concerns raised by the reviewer.*

Comment 2: 1) Google has recently made available their self-supervised large foundation model for digital pathology, that is now available for use to the public through an API, hence why not use that for the feature embedding and then just use an interpretable mechanistic lightweight model for the downstream task at hand?

blog: <https://blog.research.google/2024/03/health-specific-embedding-tools-for.html>

paper: <https://arxiv.org/abs/2310.13259>

code API:

https://docs.google.com/forms/d/1auyo2VklzuiAXavZy1AWUyQHAqO7T3BLK-7ofKUvug/viewform?edit_requested=true#question=1168037695&field=173852432

Response: *The reviewer makes a good point and we agree that in the future, foundation models can potentially be used as a first step in similar studies. Recently, foundation models, including our own HP-Atlas¹, have been shown to perform well in high-level downstream tasks, e.g. cancer subtyping. However, whether these pan-cancer models can identify consistent and interpretable histomorphological patterns at high resolution in any given tissue type is still unclear. This is why we chose to train a self-supervised (SSL) model specifically on colon cancer images. It would be very interesting to compare tissue-specific SSL models to pan-cancer models, but we believe it is beyond the scope of this work to do this systematically. Nevertheless, since this is a very interesting question, we decided to proceed with some initial analyses to quantify the concordance between these approaches.*

Specifically, we applied two pre-trained foundation models, UNI² and HP-atlas¹, to convert tile images to embeddings in the training set of TCGA colon adenocarcinoma (TCGA-COAD) and in the AVANT (Bevacizumab-Avastin®adjuVANT trial) external test set. Next, following a similar analysis pipeline as outlined in the manuscript, we performed Leiden clustering based on the UNI and HP-atlas features in TCGA-COAD. We selected the Leiden configurations for the two

foundation models based on the number of clusters obtained, aiming to match as closely as possible the clusters generated by our Barlow Twins-COAD model.

Then, we calculated the Adjusted Mutual Information (AMI)³ to evaluate the consistencies between COAD-BarlowTwins clusters and clusters derived from the two foundation models. The AMI between Barlow Twins-COAD and HP-Atlas-based clusters was estimated as 0.432, the AMI between Barlow Twins-COAD and UNI-based clusters was estimated as 0.387.

Additionally, we evaluated the consistency of the histologic patterns identified by the two foundation models by conducting the pathologist evaluation. In brief, the assessor was shown three groups of image tiles, each containing five tiles. Two groups were from the same HPC, and the third was from a randomly selected other HPC, also called the "odd HPC". The assessor was required to identify the "odd HPC". Each HPC underwent 50 tests to determine the success rate (see Online Methods: Pathologist assessment of HPCs, for details). The pairwise comparisons of test accuracies from the three models are shown in Fig. 1d-f. Wilcoxon signed-rank tests revealed no significant difference in test accuracy between Barlow-Twins-COAD-based and HP-atlas-based clusters ($p=0.159$). However, Barlow-Twins-COAD-based clusters demonstrated significantly better test accuracy compared to UNI-based clusters ($p=0.0217$).

Figure 1. Comparison of objective pathologist tests in BarlowTwins-based, UNI-based and HP-Atlas-based clusters. (a-c) Test results for BarlowTwins-based, UNI-based clusters

and HP-Atlas-based clusters in the AVANT study. Green bars indicate the percentage of successfully identified odd clusters. (d-f) Box plots and Wilcoxon signed-rank tests show pairwise comparisons in success rates between Barlow-Twins-COAD and HP-Atlas(d), Barlow-Twins-COAD and UNI (e), as well as HP-Atlas and UNI models (f).

Lastly, we assessed how well the clusters derived from the two foundation models predicted overall survival in the AVANT external test set. The prediction performances from the UNI-based clusters (c-index=0.60, log-rank p-value=0.022) and HP-Atlas-based (c-index=0.64, p-value=0.055) in the AVANT external test results were shown in Fig. 2a and b. Our conclusion is that HP-Atlas and UNI foundation models do not outperform the overall survival (OS) prediction performance achieved from the Barlow Twins-COAD HPC (c-index = 0.65, p-value = 0.005).

Figure 2. OS prediction performance in the external AVANT test set. (a) UNI-based OS prediction; (b) HP-Atlas-based OS prediction.

Comment 3: As a computer scientist, I am not completely convinced regarding the choice of Barlow Twins model for this application, especially since Barlow Twins was proposed to be specifically INVARIANT to certain feature artifacts... though this is helpful in structured applications to avoid redundant features, but cancer is so heterogeneous! How do you know that some specific feature to a certain tumour sub-type would not be treated as outlier, and hence detract from personalized predictions?

Response: We appreciate the reviewer's comment about invariance to certain feature artifacts, we believe this is an interesting and important point to clarify.

Before doing so, in order to avoid any confusion, we want to first clarify that the invariance property of Barlow Twins is not used to "avoid redundant features". In fact, the loss function has two terms, one for invariance and a different one for redundancy. It seems that this comment is about invariance and the next comment is about redundancy, so we will first address the invariance comment here.

As the reviewer points out, an invariance term is used in the loss function to favor embeddings that are invariant to certain artifacts. The reviewer raises an interesting point, wondering whether enforcing invariance may eliminate features that are histologically/biologically meaningful in heterogeneous tumors. To address this concern, we first point out that our model enforces invariance to properties that are not relevant to cancer heterogeneity. Specifically, we *applied standard crop, rotation, flip, and color distortion as suggested by Barlow Twins⁵, which are considered to be irrelevant to core features of histopathology images. Indeed, similar data augmentations, including color distortion, have been widely used in other networks, such as DINO-v2 ⁴ applied in the UNI foundation model. Furthermore, to address any remaining doubts, we reference our recent work where we used Barlow Twins to map the landscape of histomorphological phenotypes in lung adenocarcinoma¹. In this study, we showed that using the embeddings from Barlow Twins, we can derive clusters of histomorphological phenotypes that recapitulate all known simple and complex histologic patterns in lung adenocarcinoma (LUAD), suggesting that the full spectrum of heterogeneity is captured even within a specific cancer sub-type (i.e. LUAD).*

Comment 4: Again, connected to the last point, a statistical evaluation of the feature invariance and its clinical alignment is needed... for example is it possible to tweak the Barlow Twins to leak some features and then note whether those were actually redundant or not?

Response: *It seems that this comment is related to redundancy and not invariance. The redundancy term in the loss function encourages the learned features to be decorrelated, meaning that different features should capture different aspects of the data. The redundancy term is weighted by the "lambda" coefficient, with a smaller lambda leading to models learning features with higher redundancy. Initially, we set the "lambda" value to 0.05 when training the model as recommended in the Barlow Twins paper⁵. Following the reviewer's suggestion, we experimented with "extreme" values of the "lambda" weights to explore the impact of allowing more redundancy in the learned features (Fig 3). To this end, we retrained our models from scratch with new lambda values and repeated the entire computational analyses. We hypothesized that increased feature redundancy would result in fewer histomorphological phenotype clusters (HPCs) and potentially capture fewer meaningful histopathological structures. In line with our hypothesis, we indeed observed fewer clusters from the lambda-0.001 (N=42) and lambda-0.00001 (N=41) models as compared to the original lambda-0.05 model (N=47) (Fig. 3). This analysis suggests that when lambda is decreased, and as a result the number of redundant (i.e. highly correlated) features is increased, there is an impact, as expected, on the number of identified histologic patterns (HPCs). This observation highlights the usefulness of the redundancy term.*

Figure 3. Experiments with various lambda values for feature redundancy. (a) Relationship between lambda hyperparameter values and number of resulting HPCs. (b) OS prediction from lambda=0.001 model in the external AVANT cohort. (c) OS prediction from lambda=0.00001 model in the external AVANT cohort

Reviewer #1 (Remarks on code availability):

Comment 5: I have reviewed the code in the sense that it provides solid documentation as well as visualisation of the results on github itself even without installing and running the code itself locally, which I did not have time to do.

Response: We thank the reviewer for the positive comments on code documentation and the visualization of the results.

Reviewer #2 - ML, image analysis:

Comment 1: The authors present an application of their previously published work, histomorphological phenotype clustering (HPC) to Colon Adenocarcinoma datasets.

The authors show that by leveraging self-supervised learning, it is possible to learning phenotypically distinct clusters of image tiles and then use these clusters in downstream analysis to address specific biological questions, such as links to overall survival (OS).

The manuscript is well written and the overall approach has been shown to work. I have a few questions I would like to see addressed:

Response: *We would like to first thank the reviewer who has put time and effort in reviewing our study. We appreciate the recognition of our work and intentions, and are thankful for the compliment on how our manuscript is written. Next, we address each of the reviewer's questions below.*

Comment 2: Scholarship: There are more self-supervised approaches and applications that more closely resemble the authors work which have been omitted. Specifically Cisternino et al., 2023 - <https://www.biorxiv.org/content/10.1101/2023.08.22.554251v1.full>

Response: *As it is indeed a quickly developing field, we are grateful for this suggested reference and added it in our introduction. Moreover, we have extended our literature search to include more recent work published on self-supervised approaches and applications in our manuscript.*

Comment 3: The HPCs are mapped back onto the WSI and demonstrate significant structure and clustering, as one would expect if they represent key histological concepts. However, the authors do not assess quantitatively and robustly how well a given HPC recovers the entire structure from a WSI. I think this is important to address, because otherwise, there's no way of seeing how much is missed or mis-classified. To do this, I would suggest getting full ground truth annotations of tissue substructures and assessing the accuracy/IoY of tiles belonging to a given HPC, recovering that substructure. This would give weight to HPC proportions being truly meaningful.

Response: *The reviewer proposes a comparison to ground-truth annotations. To this end, we randomly selected 50 whole-slide images from the external dataset (AVANT) and annotated them by researcher MP manually in detail using QuPath software version 0.1.2. Annotations were drawn using standard pathologist labels based on the following tissue types: healthy and dysplastic tissue, necrosis, mucinous, muscle tissue, immune cells, fatty tissue, tumor stroma, and tumor epithelium. Annotations were checked by pathologists afterwards. We then assessed the performance of HPCs in terms of identifying correctly these manual annotations as recommended by the reviewer. First, annotation masks were generated for each manual annotation label and overlapped at the tile level on each of the 50 slides. An annotation label was assigned to a tile if at least 50% of the tile area was covered by the corresponding annotation. We then compared the HPC super-cluster tile labels with ground-truth tile labels ($N = 7,874$ tiles) and calculated a macro-AUC of 0.830 (95% bootstrap confidence intervals [CI] = 0.826-0.834, with 1000 iterations) and a micro-AUC of 0.931 (95% bootstrap CI: 0.927-0.935).*

To illustrate the excellent concordance between SSL-assigned labels and manual annotations, we plotted three randomly selected example heatmaps of pathologist tissue annotations and corresponding predicted labels formed by HPCs (Figure 4). The tissue labels predicted by HPCs (right) consistently reflect the pathologist annotated tissue areas in H&E-stained tissue slide (left).

Figure 4. HPC-based tissue areas (right) versus pathologist annotated tissue areas on H&E-stained tissue slide (left) in three separate colon carcinoma cases (A-C).

Comment 4: Many HPCs correlate / align with known risk factors for cancer, such as immune cell fraction. Are there any HPCs that have effects of OS that are truly novel and don't correlate with currently known risk factors. I understand the authors have demonstrated overall that HPCs add predictive power over baseline measures, but the interpretation is key.

Response: *We greatly appreciate the reviewer's recognition of our work on correlating HPCs with cancer risk factors and acknowledging that our HPCs provide additional predictive potential beyond a clinical baseline model (trained on age, sex, tumor-stroma ratio (TSR), and AJCC stage; see Supplementary Figure 3C).*

To directly address the reviewer's point regarding interpretations, we have again delved into the literature and had more extensive discussions with expert pathologists in trying to ascertain potential explanations of this effect on patient-related outcomes per HPC, as we highlight in 'Results' but predominantly expand on in-depth in the 'Discussion'.

We have therefore created a whole new table, Supplementary Table 2, linking HPCs to major immune and oncogenic pathways and interpretations how these associations could influence patient survival as well as their response to the experimental treatment with Bevacizumab and standard chemotherapy. As the reviewer points out, this was done by referring to current literature and knowledge. We identified certain morphologies that are not typically analyzed in pathology, such as the amount or organization of tumor stroma (e.g., HPC 0, 40), neovascularization (HPC 11, 21), and tissue composition likely corresponding to microsatellite instability status (HPC 26, 46). Additionally, we found novel characteristics that have not yet been well-described, such as necrosis and VEGFa gene enrichment (HPC 5, 18) in relation to response to Bevacizumab. To validate whether these novel markers are truly linked to novel tumor biology, future studies involving experimental studies on molecular, cellular, and animal models as well as validations on population-based studies are warranted.

Comment 5: Figure 5:

- Infiltrated stroma / moderately differentiated tumour are associated with better survival
- Longitudinal muscle fibres or vessel-like stroma are associated with poor survival?

How do I interpret this, because those are counterintuitive to me. Surely normal tissue histology and the plane it was cut in have no bearing on survival and certainly not a negative one.

Response: *First, we would like to thank the reviewer for their effort to interpret the sometimes seemingly counterintuitive findings. In response to this comment and to the previous question, we have created the new Supplementary Table 2, which includes all HPCs that are associated with OS either in the experimental treatment or standard treatment groups (as shown in manuscript Figure 5). In addition to our Supplementary Table 1 where we described tissue*

composition of each HPCs, Supplementary Table 2 specifically highlights and summarizes features and patterns in those HPCs that have been previously linked to OS in the literature (references included in table). Additionally, we have revised the Discussion section to include these findings and added extra references. Below, we have added the specifically mentioned associations the reviewer requested clarification of.

As the reviewer points out, our model indicates that the HPC labels of "Infiltrated stroma / moderately differentiated tumour are associated with better survival". First, to clarify, the labels of the HPCs are short titles, summarizing the main feature shown in that HPC, facilitating use of all HPCs in general (e.g. in figures), however, there are often more features which potentially associate with OS than only those used in the label description. With 'infiltrated' stroma, we meant 'infiltrated by immune cells, or a high influx of immune cells', for instance like HPC 13. This is a sign of a good host response against the tumor, which is a favorable outcome and there are many biomarkers known that correspond with this (e.g. Immunoscore). The well-moderately differentiated tumor epithelium (also known as 'low grade' tumor) is, in comparison to poorly-undifferentiated tumor (or high grade), also a known prognostic favorable factor, even more so with a combination of favorable factors, like illustrated by HPC 4. This is supported by the explanations and references we present in our novel Supplementary Table 2.

The reviewer also points out that our model indicates that "Longitudinal muscle fibres or vessel-like stroma are associated with poor survival". A possible explanation for the longitudinal muscle fibers, is that these also are very similar to stromal strands, indicating higher proportions of tumor stroma. Another possibility is for HPC 33 for instance, that we saw tumor buds invading the muscle tissue and/or serosa of the colon, a sign of a more aggressive tumor. Since the majority of the tiles consist of muscle fibers, the HPC is merely labeled 'Muscle tissue (longitudinal fibers)', although the association on OS is poor, which can indeed seem counterintuitive. Furthermore, vessels in stroma often indicate neovascularization, an active tumor remodeling its microenvironment and signaling for more nutrients to maximize potential invasion. The 'Vessel-like stroma/muscle' HPC 24, on the other hand, is a bit more difficult and shows different OS associations per treatment group. After extensive literature searching and with expert opinions, we summarize the evidence in Supplementary Table 2.

Regarding the non-cancerous tissue and a potential influence on outcome, there is an effect known as 'cancer field characterization' or 'field effect', in which the tissue surrounding the tumor also is affected by the storm of interleukins and cytokines, immune cells and active remodeling of the microenvironment⁶. Another potential explanation is that, as we model the fractional data of HPCs per slide per patient, higher fraction of non-cancerous tissue may indicate lower fraction of cancerous tissue in a certain slide, suggesting a potentially smaller tumor. Small tumor size, as also for instance stated in the TNM classification (lower T stage), is a known prognostic factor favoring survival.

Finally, we would like to refer the reviewer to our novel, detailed overview of the associations to OS per HPC in Supplementary Table 2, where these explanations are summarized, including references.

Reviewer #2 (Remarks on code availability):

Comment 6: The code is poorly structured, with lots of .py scripts sitting in the main repo and not organised. They provide a comprehensive README.md.

Response: *We appreciate this feedback and are grateful that the reviewer took the time to look through our codebase. We have provided a comprehensive and structured README file (<https://github.com/AdalbertoCq/Histomorphological-Phenotype-Learning/blob/master/README.md>) detailing each step required to execute the models and scripts.*

Reviewer #3 - Colon cancer, histopathology:

Comments 1: The present manuscript entitled "Self-Supervised Learning Reveals Clinically Relevant Histomorphological Patterns for Therapeutic Strategies in Colon Cancer" by Liu et al. is a well-written and concisely presented study.

1. Strengths of the study
 - a) Interdisciplinary team
 - b) Test set and validation set
 - c) Excellent visualisation of the study results
 - d) Figure 7 highlights the potential clinical implication very well

Response: *We are thankful for all the compliments and the recognition of the strengths of our work by the reviewer.*

Comments 2: 2. Limitations of the study

- a. My major points are already addressed by the authors in the discussion
- b. In my opinion an additional paragraph should be added at the end of the discussion trying to address the following points:
 - How should Institutes of Pathology concretely implement this approach?
 - First report traditionally, then apply the proposed method in terms of additional information?
 - Or shall this approach replace the standard reporting?
 - Do the authors plan a validation study including more centers?
 - How can the obtained results be correlated with the actual parameters proposed by the ICCR (International Collaboration on Cancer Reporting)?

Response: We thank the reviewer for acknowledging that our Discussion already addressed the major points. To address the reviewer's comments about potential clinical implementation and future validations, we have now indeed revised the Discussion and included an additional paragraph as suggested by the reviewer, as here below:

"Building upon the aforementioned findings, this study showcases the prospective clinical utility of AI-generated HPCs (Figure 7). Cancer WSIs were preprocessed into image patches and subsequently used to train SSL encoders and to form HPCs. These HPCs serve as condensed representations of the original WSIs, ready to be inspected by pathologists and enabling flexible linkage to multimodal omics data. These HPCs hold promise in classifying various tumor characteristics, potentially predicting patient prognosis and discerning distinct sensitivity groups to various therapies. Although this study was already trained on multicenter TCGA data and validated in an external, clinical multicenter cohort, we plan to conduct additional validations in other population-based external cohorts to strengthen its clinical applicability. In a pathology lab, this could be implemented as follows: Alongside the routine pathology report, an AI-generated report would provide personalized prognostic risk quantification (e.g., based on the patient's HPC composition and SHAP values). The report would also include tissue composition descriptions for each HPC, granting pathologists a complete overview per patient. Subsequently, with these reports used in the multidisciplinary team meetings, a colon cancer patient can be granted an optimal personalized treatment strategy."

Regarding the actual correlation to the mentioned ICCR parameters, we would first like to refer the reviewer to our Supplementary Table 1, where we have described all HPCs in detail. We will highlight some here, of which we have also added some to our Discussion:

Given the significant potential implications of these HPCs on OS and therapy response, implementation of this AI-based analysis may be advocated to international guideline organizations, such as the TNM evaluation committee of the Union for International Cancer Control (UICC). Our HPC-based analysis not only summarizes the TNM classification, but also correlates to parameters assessed in standard pathology diagnostics, such as the International Collaboration on Cancer Reporting (ICCR)⁷. Of note, ICCR parameters pertain to a global assessment on whole slide or patient level, while HPCs can provide a local assessment of histopathological characteristics on the tile level. However, these tile level histopathological characteristics can still be linked to global ICCR parameters. For example, a core element of the ICCR, 'Histological tumor grade', can be correlated to HPCs containing tumor epithelium with different differentiation statuses. HPCs 2 and 3 are for instance characterized by well-moderately differentiated (or 'low-grade') tumor epithelium, while HPC 45 is formed by poorly-to-undifferentiated (or 'high-grade') tumor epithelium-containing tiles. Moreover, 'Histological tumor type' in the ICCR can be linked to HPC 12, which contains predominantly mucinous tumors. Lastly, the TNM classification, and particularly tumor size, in the ICCR reporting guidelines may be linked to HPCs containing healthy colon tissue (e.g. HPC 39), since WSIs with more of the healthy HPC 39, may correspond to a fractionally smaller tumor size (i.e. lower T-stage)."

Response to editor and reviewers - Liu et al. 'Self-Supervised Learning Reveals Clinically Relevant Histomorphological Patterns for Therapeutic Strategies in Colon Cancer' (**NCOMMS-24-11291A**)

References

1. Claudio Quiros et al. Mapping the landscape of histomorphological cancer phenotypes using self-supervised learning on unannotated pathology slides. Nat. Commun. 2024
2. Chen et al. Towards a general-purpose foundation model for computational pathology. Nat. Med. 2024
3. Romano et al. Adjusting for Chance Clustering Comparison Measures. J. Mach. Learn. Res. 2016
4. UNI/uni at main · mahmoodlab/UNI. GitHub
5. Zbontar et al. Barlow Twins: Self-Supervised Learning via Redundancy Reduction. BioXriv 2021.
6. Lockhead et al. Etiologic field effect: reappraisal of the field effect concept in cancer predisposition and progression. Mod. Path. 2015
7. Loughrey et al. Colorectal Cancer Histopathology Reporting Guide. 1st edition. International Collaboration on Cancer Reporting; Sydney, Australia. 2020. ISBN: 978-1-922324-01-6.